# Fast Debiasing of the LASSO Estimator

**Shuvayan Banerjee**  *banerjeeshuvayan21@gmail.com*
*Department of Mathematics*
*IITB–Monash Research Academy, Mumbai, India*

**James Saunderson**  *james.saunderson@monash.edu*
*Department of Electrical and Computer Systems Engineering*
*Monash University, Melbourne, Australia*

**Radhendushka Srivastava**  *rsrivastava@iitb.ac.in*
*Department of Mathematics*
*IIT Bombay, Mumbai, India*

**Ajit Rajwade**  *ajitvr@cse.iitb.ac.in*
*Department of Computer Science and Engineering*
*IIT Bombay, Mumbai, India*

**Reviewed on OpenReview:** *https://openreview.net/forum?id=gEVPlLhoNI*

## Abstract

In high-dimensional sparse regression, the LASSO estimator offers excellent theoretical guarantees but is well-known to produce biased estimates. To address this, Javanmard & Montanari (2014a) introduced a method to "debias" the LASSO estimates for a random sub-Gaussian sensing matrix $\boldsymbol{A}$. Their approach relies on computing an "approximate inverse" $\boldsymbol{M}$ of the matrix $\boldsymbol{A}^\top \boldsymbol{A}/n$ by solving a convex optimization problem. This matrix $\boldsymbol{M}$ plays a critical role in mitigating bias and allowing for construction of confidence intervals using the debiased LASSO estimates. However the computation of $\boldsymbol{M}$ is expensive in practice as it requires iterative optimization. In the presented work, we re-parameterize the optimization problem to compute a "debiasing matrix" $\boldsymbol{W} := \boldsymbol{A}\boldsymbol{M}^\top$ directly, rather than the approximate inverse $\boldsymbol{M}$. This reformulation retains the theoretical guarantees of the debiased LASSO estimates, as they depend on the *product* $\boldsymbol{A}\boldsymbol{M}^\top$ rather than on $\boldsymbol{M}$ alone. Notably, we derive a simple and computationally efficient closed-form expression for $\boldsymbol{W}$, applicable to the sensing matrix $\boldsymbol{A}$ in the original debiasing framework, under a specific deterministic condition. This condition is satisfied with high probability for a wide class of randomly generated sensing matrices. Also, the optimization problem based on $\boldsymbol{W}$ guarantees a unique optimal solution, unlike the original formulation based on $\boldsymbol{M}$. We verify our main result with numerical simulations.

## 1 Introduction

In high-dimensional sparse regression, where the number of predictors significantly exceeds the number of observations, the LASSO (Least Absolute Shrinkage and Selection Operator) is a widely used method for variable selection and estimation. By incorporating an $\ell_1$ regularization term, LASSO promotes sparsity in the estimated coefficients, enabling effective performance for sparse signal vectors even if the number of predictors far exceeds the number of samples. The LASSO estimator has well-established theoretical guarantees for signal and support recovery (Hastie et al., 2015). Despite its strengths, a well-recognized limitation of LASSO is its tendency to produce biased estimates. This bias arises from the shrinkage imposed by the $\ell_1$ penalty. Consequently, the bias compromises estimation accuracy and impedes statistical inference tasks

such as construction of confidence intervals or hypothesis tests. These challenges are especially pronounced in high-dimensional regimes, where traditional inference tools fail due to high dimensionality.

To address these limitations, several methods have been developed to "debias" the LASSO estimator, allowing for valid statistical inference even in high-dimensional settings. Notably, Zhang & Zhang (2014) introduced a decorrelated score-based approach, leveraging the Karush–Kuhn–Tucker (KKT) conditions of the LASSO optimization problem to construct bias-corrected estimators. Their framework relies on precise estimation of the precision matrix (inverse covariance matrix), which can be computationally challenging and sensitive to regularization choices. Similarly, Van de Geer et al. (2014) proposed a methodology rooted in node-wise regression, where each variable is regressed on the remaining variables to estimate the precision matrix. While effective, this method is computationally intensive. This may limit its applicability, particularly in scenarios where the design matrix lacks favorable properties like sparsity of the rows of the precision matrix.

Javanmard & Montanari (2014a) introduced a simple yet powerful approach that constructs debiased LASSO estimates using an "approximate inverse" of the sample covariance matrix. Their method avoids direct precision matrix estimation and instead employs an optimization framework to compute a debiasing matrix $\boldsymbol{M}$ that corrects for bias while ensuring asymptotic normality of the debiased estimates. A key advantage of this method is its applicability for random sub-Gaussian sensing matrices, enabling valid inference across a broad range of high-dimensional applications.

In this work, we build upon the technique of Javanmard & Montanari (2014a), addressing one of its primary computational bottlenecks: the optimization step required to compute the approximate inverse $\boldsymbol{M}$. By reformulating the problem to work directly with the "weight matrix" $\boldsymbol{W} := \boldsymbol{A}\boldsymbol{M}^\top$, we entirely eliminate the need to solve this optimization problem in many practical cases. Our proposed reformulation leverages the insight that the theoretical guarantees of the debiased LASSO estimator depend on the product $\boldsymbol{A}\boldsymbol{M}^\top$ rather than the individual debiasing matrix $\boldsymbol{M}$. By shifting the focus to the "weight matrix" $\boldsymbol{W} := \boldsymbol{A}\boldsymbol{M}^\top$, we simplify the optimization problem while retaining all theoretical properties of the original framework. Under certain deterministic assumptions, we provide a simple, exact, closed form optimal solution for the optimization problem to obtain $\boldsymbol{W}$. We show that this assumption is satisfied with high probability for different popular ensembles of sub-Gaussian sensing matrices, under the additional condition that the elements of the rows of $\boldsymbol{A}$ are weakly correlated. In practice, sensing matrices with uncorrelated entries are commonly used in many applications (Duarte et al., 2008; Liu et al., 2013) and are also widely used in many theoretical results in sparse regression (Hastie et al., 2015). This closed form solution eliminates the computationally intensive optimization step required to compute $\boldsymbol{M}$, significantly improving runtime efficiency. It is applicable in many natural situations, including sensing matrices with i.i.d. isotropic sub-Gaussian rows (such as i.i.d. Gaussian, or i.i.d. Rademacher entries).

**Notation:** Throughout this paper, we denote matrices by bold-faced uppercase symbols, e.g., $\boldsymbol{A}$. If $\boldsymbol{A}$ is an $n \times p$ matrix then $\boldsymbol{a}_{i\cdot} \in \mathbb{R}^p$ denotes the $i^{\text{th}}$ row of $\boldsymbol{A}$, thought of as a column vector. Similarly if $\boldsymbol{A}$ is an $n \times p$ matrix then $\boldsymbol{a}_{\cdot j} \in \mathbb{R}^n$ denotes the $j^{\text{th}}$ column of $\boldsymbol{A}$, again thought of as a column vector. Vectors are denoted by bold-faced lower case symbols, e.g., $\boldsymbol{w}$. The $i$th entry of a vector $\boldsymbol{w}$ is denoted $w_i \in \mathbb{R}$. The identity matrix of size $p \times p$ for any positive integer $p$ is denoted by $\boldsymbol{I_p}$, and its $i^{\text{th}}$ column vector is denoted by $\boldsymbol{e}_i$. For a positive integer $p$, we use the shorthand $[p] = \{1, 2, \ldots, p\}$. For a vector $\boldsymbol{w} \in \mathbb{R}^m$, we denote the $\ell_q$-norm by $\|\boldsymbol{w}\|_q := \left(\sum_{i=1}^m |w_i|^q\right)^{1/q}$ if $1 \leq q < \infty$ and the $\ell_\infty$-norm by $\|\boldsymbol{w}\|_\infty := \max_{i \in [m]} |w_i|$.

## 2 An Overview of the Debiased LASSO

We consider the high-dimensional linear model

$$\boldsymbol{y} = \boldsymbol{A}\boldsymbol{\beta^*} + \boldsymbol{\eta}, \tag{1}$$

where $\boldsymbol{\beta^*} \in \mathbb{R}^p$ is a $s$-sparse signal (i.e., $s := \|\boldsymbol{\beta^*}\|_0$ where $s \ll p$), $\boldsymbol{A}$ is a $n \times p$ design/sensing matrix (where $n \ll p$), and $\boldsymbol{y} \in \mathbb{R}^n$ is the measurement vector. Also, $\boldsymbol{\eta} \in \mathbb{R}^n$ is an additive noise vector that consists of independent and identically distributed elements drawn from $\mathcal{N}(0, \sigma^2)$, where $\sigma^2$ is the noise variance.

The LASSO estimate $\hat{\boldsymbol{\beta}}_{\boldsymbol{\lambda}}$ of the sparse signal $\boldsymbol{\beta}^*$ is defined as the solution to the following optimization problem:

$$\hat{\boldsymbol{\beta}}_{\boldsymbol{\lambda}} := \arg\min_{\boldsymbol{\beta}} \frac{1}{2n}\|\boldsymbol{y} - \boldsymbol{A}\boldsymbol{\beta}\|_2^2 + \lambda\|\boldsymbol{\beta}\|_1, \tag{2}$$

where $\lambda > 0$ is a regularization parameter chosen appropriately. The LASSO estimator is known to be a consistent estimator of the sparse signal $\boldsymbol{\beta}^*$ under the condition that the sensing matrix $\boldsymbol{A}$ satisfies the Restricted Eigenvalue Condition (REC) (Hastie et al., 2015, Chapter 11).

The LASSO estimator is well-known to produce biased estimates, i.e., $E(\hat{\boldsymbol{\beta}}_{\boldsymbol{\lambda}}) \neq \boldsymbol{\beta}^*$ where the expectation is computed over noise instances. This bias arises from the $\ell_1$ regularization term, which induces shrinkage in the estimate $\hat{\boldsymbol{\beta}}_{\boldsymbol{\lambda}}$. Moreover, there is no known method to compute a confidence interval of $\boldsymbol{\beta}^*$ directly from $\hat{\boldsymbol{\beta}}_{\boldsymbol{\lambda}}$.

To reduce this bias and also construct confidence intervals of $\boldsymbol{\beta}^*$, Javanmard & Montanari (2014a) introduced a debiased LASSO estimator $\hat{\boldsymbol{\beta}}_{\boldsymbol{d}}$, defined as follows:

$$\hat{\boldsymbol{\beta}}_{\boldsymbol{d}} := \hat{\boldsymbol{\beta}}_{\boldsymbol{\lambda}} + \frac{1}{n}\boldsymbol{M}\boldsymbol{A}^\top(\boldsymbol{y} - \boldsymbol{A}\hat{\boldsymbol{\beta}}_{\boldsymbol{\lambda}}). \tag{3}$$

Here $\boldsymbol{M}$ is an approximate inverse of the rank deficient matrix $\hat{\boldsymbol{\Sigma}} := \boldsymbol{A}^\top\boldsymbol{A}/n$, computed by solving the convex optimization problem given in Algorithm 1. The parameter $\mu$ in Alg.1 controls the bias of the debiased LASSO estimator given in (3). Ideally one should choose the smallest $\mu$ for which (4) is feasible.

---

**Algorithm 1** Construction of $\boldsymbol{M}$ (from Javanmard & Montanari (2014a))

---

**Require:** Design matrix $\boldsymbol{A}$, $\mu \in (0,1)$
**Ensure:** Debiasing matrix $\boldsymbol{M}$
1: Compute: $\hat{\boldsymbol{\Sigma}} := \boldsymbol{A}^\top\boldsymbol{A}/n$.
2: For each $j \in [p]$, solve the following optimization problem to compute column vector $\boldsymbol{m}_{\cdot j} \in \mathbb{R}^p$:

$$\text{minimize} \quad \boldsymbol{m}_{\cdot j}^\top \hat{\boldsymbol{\Sigma}} \boldsymbol{m}_{\cdot j}$$
$$\text{subject to} \quad \|\hat{\boldsymbol{\Sigma}}\boldsymbol{m}_{\cdot j} - \boldsymbol{e}_j\|_\infty \leq \mu, \tag{4}$$

where $\boldsymbol{e}_j$ is the $j^{\text{th}}$ column of the identity matrix $\boldsymbol{I_p}$, and $\mu \in (0,1)$.
3: Assemble $\boldsymbol{M}$ as $\boldsymbol{M} := (\boldsymbol{m}_{\cdot 1}|\cdots|\boldsymbol{m}_{\cdot p})^\top$.
4: If the optimization problem is infeasible for any $j$, set $\boldsymbol{M} := \boldsymbol{I_p}$.

---

The theoretical properties of $\hat{\boldsymbol{\beta}}_{\boldsymbol{d}}$ are applicable to a sensing matrix $\boldsymbol{A}$ with the following properties:

**D1:** The rows $\boldsymbol{a}_{1\cdot}, \boldsymbol{a}_{2\cdot}, \ldots, \boldsymbol{a}_{n,\cdot}$ of matrix $\boldsymbol{A}$ are independent and identically distributed zero-mean sub-Gaussian random vectors with covariance $\boldsymbol{\Sigma} := E[\boldsymbol{a}_{i\cdot}\boldsymbol{a}_{i\cdot}^\top]$. Furthermore, the sub-Gaussian norm $\kappa := \|\boldsymbol{\Sigma}^{-1/2}\boldsymbol{a}_{i\cdot}\|_{\psi_2}$ [1] is a finite positive constant.

**D2:** There exist positive constants $0 < C_{\min} \leq C_{\max}$, such that the minimum and maximum eigenvalues $\sigma_{\min}(\boldsymbol{\Sigma})$, $\sigma_{\max}(\boldsymbol{\Sigma})$ of $\boldsymbol{\Sigma}$ satisfy $0 < C_{\min} \leq \sigma_{\min}(\boldsymbol{\Sigma}) \leq \sigma_{\max}(\boldsymbol{\Sigma}) \leq C_{\max} < \infty$.

Theorem 7(b) of Javanmard & Montanari (2014a) shows that the optimization problem in (4) is feasible with high probability, for sensing matrices satisfying properties **D1** and **D2**, as long as $\mu > 4\sqrt{3}e\kappa^2\sqrt{\frac{C_{\max}}{C_{\min}}}\sqrt{\frac{\log p}{n}}$. If $\mu$ is $O\left(\sqrt{\frac{\log p}{n}}\right)$ and $n$ is $\omega((s\log p)^2)$, then Theorem 8 in the aforementioned paper shows that the bias of the debiased LASSO estimator goes to 0 and $\forall j \in [p]$, $\sqrt{n}(\hat{\beta}_{dj} - \beta_j^*)$ is asymptotically zero-mean Gaussian with variance $\sigma^2 \boldsymbol{m}_{\cdot j}^\top \hat{\boldsymbol{\Sigma}} \boldsymbol{m}_{\cdot j}$.

---

[1]The sub-Gaussian norm of a random variable $x$, denoted by $\|x\|_{\psi_2}$, is defined as $\|x\|_{\psi_2} := \sup_{q\geq 1} q^{-1/2}\left(E|x|^q\right)^{1/q}$. For a random vector $\boldsymbol{x} \in \mathbb{R}^n$, its sub-Gaussian norm is defined as $\|\boldsymbol{x}\|_{\psi_2} := \sup_{\boldsymbol{y}\in S^{n-1}} \|\boldsymbol{y}^\top\boldsymbol{x}\|_{\psi_2}$, where $S^{n-1}$ denotes the unit sphere in $\mathbb{R}^n$.

# 3 Re-parameterization of the Debiased LASSO

The debiased LASSO estimator in (3) can be rewritten in terms of the weight matrix $\boldsymbol{W} := \boldsymbol{A}\boldsymbol{M}^\top$ as:

$$\hat{\boldsymbol{\beta}}_{\boldsymbol{d}} = \hat{\boldsymbol{\beta}}_{\boldsymbol{\lambda}} + \frac{1}{n}\boldsymbol{W}^\top(\boldsymbol{y} - \boldsymbol{A}\hat{\boldsymbol{\beta}}_{\boldsymbol{\lambda}}). \tag{5}$$

The re-parameterization does not affect the debiasing procedure described earlier. Thus, any theoretical guarantees established using $\boldsymbol{M}$ extend to those using $\boldsymbol{W}$.

We now produce a reformulated problem in (6) using $\boldsymbol{W}$, and show that it is equivalent to the original optimization problem in Algorithm 1. Using the relationship $\boldsymbol{W} = \boldsymbol{A}\boldsymbol{M}^\top$, we can rewrite $\boldsymbol{m}_{.j}$ as $\boldsymbol{w}_{.j} := \boldsymbol{A}\boldsymbol{m}_{.j}$. Making this substitution, the objective in (4) becomes $\boldsymbol{m}_{.j}^\top\hat{\boldsymbol{\Sigma}}\boldsymbol{m}_{.j} = \frac{1}{n}\boldsymbol{w}_{.j}^\top\boldsymbol{w}_{.j}$ and the constraint $\|\hat{\boldsymbol{\Sigma}}\boldsymbol{m}_{.j} - \boldsymbol{e}_j\|_\infty \le \mu$ (where $\boldsymbol{e}_j$ is the $j$th column of the identity matrix) becomes $\left\|\frac{1}{n}\boldsymbol{A}^\top\boldsymbol{w}_{.j} - \boldsymbol{e}_j\right\|_\infty \le \mu$. This change of variables suggests the following reformulated optimization problem (6) for the $j^{\text{th}}$ column of $\boldsymbol{W}$:

$$\begin{aligned}\mathscr{P}_j := \text{minimize} \quad & \tfrac{1}{n}\boldsymbol{w}_{.j}^\top\boldsymbol{w}_{.j} \\ \text{subject to} \quad & \left\|\tfrac{1}{n}\boldsymbol{A}^\top\boldsymbol{w}_{.j} - \boldsymbol{e}_{\boldsymbol{j}}\right\|_\infty \le \mu.\end{aligned} \tag{6}$$

In fact, the $j^{\text{th}}$ reformulated problem (6) and the $j^{\text{th}}$ original problem (4) are equivalent in the following sense: If $\boldsymbol{m}_{.j}$ is feasible for (4) then $\boldsymbol{w}_{.j} := \boldsymbol{A}\boldsymbol{m}_{.j}$ is feasible for (6) and $\frac{1}{n}\boldsymbol{w}_{.j}^\top\boldsymbol{w}_{.j} = \boldsymbol{m}_{.j}^\top\hat{\boldsymbol{\Sigma}}\boldsymbol{m}_{.j}$. Conversely, suppose that $\boldsymbol{w}_{.j}$ is feasible for (6). If $\boldsymbol{A}^\dagger$ is a pseudo-inverse of $\boldsymbol{A}$, then $\boldsymbol{m}_{.j} := \boldsymbol{A}^\dagger\boldsymbol{w}_{.j}$ is feasible for (4) since $\hat{\boldsymbol{\Sigma}}\boldsymbol{m}_{.j} = \frac{1}{n}\boldsymbol{A}^\top\boldsymbol{A}\boldsymbol{m}_{.j} = \frac{1}{n}\boldsymbol{A}^\top\boldsymbol{w}_{.j}$. Moreover, $\frac{1}{n}\boldsymbol{w}_{.j}^\top\boldsymbol{w}_{.j} = \boldsymbol{m}_{.j}^\top\hat{\boldsymbol{\Sigma}}\boldsymbol{m}_{.j}$, so both have the same objective values, establishing that (4) and (6) are equivalent. This reformulation provides an equivalent separable problem for each column of $\boldsymbol{W}$, maintaining all theoretical guarantees while simplifying the representation of the debiasing procedure.

The reformulated problem (6) has a *unique* optimal solution because the objective function is strongly convex with convex constraints. In contrast, the original problem (4) does not have a unique solution. Indeed if $\boldsymbol{m}_{.j}$ is any solution to (4), then we can add to it any element of the nullspace of $\boldsymbol{A}$ to obtain another solution to (4).

## 3.1 A Closed-Form Solution for the Debiasing Matrix $\boldsymbol{W}$

In this section, we demonstrate that, for a suitable choice of $\mu$, the optimal solution to the problem (6) can be computed in closed form for a sensing matrix whose minimum column norm is strictly positive (which is true with probability 1 for random matrices). To derive this result we write down the Fenchel dual of (6), and appeal to weak duality. In particular, we explicitly find primal and dual feasible points with the same objective value, certifying that both are, in fact, optimal.

**Theorem 1** *Let $\boldsymbol{A}$ be a $n \times p$ matrix with no column equal to zero. Define $\rho(\boldsymbol{A}) := \max_{i \ne j} \frac{|\boldsymbol{a}_{.i}^\top\boldsymbol{a}_{.j}|}{\|\boldsymbol{a}_{.j}\|_2^2}$. The optimal solution of (6) is given by*

$$\boldsymbol{w}_{.j} := \frac{n(1 - \mu)}{\|\boldsymbol{a}_{.\boldsymbol{j}}\|_2^2}\boldsymbol{a}_{.j} \quad \text{for all } j \in [p] \tag{7}$$

*if and only if $\frac{\rho}{1+\rho} \le \mu \le 1$.*

The proof of this theorem is given in Appendix A.1. For notational simplicity, we will denote $\rho(\boldsymbol{A})$ by $\rho$ in the rest of the paper. We will provide a brief overview of the proof here.

**Overview of the proof of Theorem 1:** The proof shows that the closed-form expression in (7) is optimal for the convex program (6) when $\frac{\rho}{1+\rho} \le \mu \le 1$. Under this condition, the candidate vector satisfies the $\ell_\infty$ feasibility constraint given in (6) because the coherence bound ensures $(1 - \mu)\rho \le \mu$. Its objective value can

be computed in closed form and is given by $\frac{(1-\mu)^2}{\|\boldsymbol{a}_{\cdot j}\|_2^2/n}$. Using a Fenchel dual reformulation, the dual problem is explicitly provided, and a suitably chosen dual vector attains a dual value that matches the primal value achieved by (7). By weak duality, this equality certifies optimality. Conversely, if $\mu < \rho/(1+\rho)$, the solution in (7) violates the feasibility constraint, and when $\mu > 1$, the zero vector is the unique minimizer which is trivially optimal. Therefore, the stated range of $\mu$ is necessary and sufficient for the optimality of the exact solution.

**Remarks:**

1. This theorem eliminates the requirement to execute an iterative optimization algorithm to obtain $\boldsymbol{W}$ (or an iterative optimization algorithm to obtain $\boldsymbol{M}$). This is because given $\boldsymbol{A}$, one can directly implement the optimal solution of Alg. 1 in the form (7) for all $j \in [p]$. This speeds up the implementation of the debiasing of LASSO for the ensemble of sensing matrices that satisfy the conditions of Theorem 1. Likewise, our approach will also be significantly faster than the debiasing approach presented in Van de Geer et al. (2014), which explicitly estimates the precision matrix (inverse covariance matrix) of the rows of $\boldsymbol{A}$ via a series of $p$ different LASSO problems, each solved iteratively – see equations 7,8,9 of Van de Geer et al. (2014).

2. The condition $\rho/(1+\rho) \le \mu \le 1$ is necessary and sufficient for the closed-form expression in (7) to be optimal (6). However, it is possible that (6) is feasible for values of $\mu$ that are smaller than $\rho/(1+\rho)$. In such a situation, the optimal solution to (6) is not given by (7). This is empirically illustrated in Sec. 5.1. In the context of the debiased LASSO, Theorem 7(b) of Javanmard & Montanari (2014b) shows that the choice of $\mu := O_P(\sqrt{\log p/n})$ makes the optimization problem in (4) feasible for a sensing matrix $\boldsymbol{A}$ that satisfies conditions **D1** and **D2**. For a wide class of sensing matrices, we show in Theorem 2 that $\rho/(1+\rho) = O_P(\sqrt{\log p/n})$.

3. The quantity $\frac{\rho}{1+\rho}$ can be computed exactly by using the definition of $\rho$ given in Theorem 1 given a sensing matrix $\boldsymbol{A}$. Furthermore, the distribution of $\frac{\rho}{1+\rho}$ can also be estimated via simulation for given any $n$, $p$, and $\boldsymbol{\Sigma}$ corresponding to the sensing matrix $\boldsymbol{A}$.

4. The solution in (7) is the optimal solution even when we choose $\mu = 1$. The optimal solution in this case is the trivial solution $\boldsymbol{w}_j = 0$. However in practice, one always chooses $\mu$ to be small, and hence this specific situation does not arise.

## 3.2 Concentration bounds of $\frac{\rho}{1+\rho}$

As mentioned earlier at the end of Sec. 2, if $\mu$ is $O\left(\sqrt{\frac{\log p}{n}}\right)$ and $n$ is $\omega((s \log p)^2)$, then $\forall j \in [p]$, $\sqrt{n}(\hat{\beta}_{dj} - \beta_j^*)$ is asymptotically zero-mean Gaussian when the elements of $\boldsymbol{\eta}$ are drawn from $\mathcal{N}(0, \sigma^2)$. For specific classes of random sensing matrices, we show in Theorem 2, that $\frac{\rho}{1+\rho} \le c_0\sqrt{\frac{\log p}{n}}$ with high probability for some constant $c_0$. This implies that for these random sensing matrices, the choice $\mu := O\left(\sqrt{\frac{\log p}{n}}\right)$ ensures *both* the following: ($i$) asymptotic negligible bias of the estimator $\hat{\boldsymbol{\beta}}_{\boldsymbol{d}}$ given by (5) when $n$ is $\omega((s \log p)^2)$, and ($ii$) fulfillment of the sufficient condition $\frac{\rho}{1+\rho} \le \mu$ for the debiasing matrix $\boldsymbol{W}$ to be computed in closed-form. If $\boldsymbol{A}$ satisfies an additional mild assumption as given in Theorem 2 then $\frac{\rho}{1+\rho} \le c_0\sqrt{\frac{\log p}{n}}$ holds with high probability.

**Theorem 2** *Let $\boldsymbol{A}$ be a $n \times p$ dimensional matrix with independent and identically distributed zero-mean sub-Gaussian rows and sub-Gaussian norm $\kappa := \|\boldsymbol{\Sigma}^{-1/2}\boldsymbol{a}_{i\cdot}\|_{\psi_2}$, where $n < p$ and $\boldsymbol{\Sigma} := E[\boldsymbol{a}_{i\cdot}\boldsymbol{a}_{i\cdot}^\top]$. Let $\rho$ be as defined in Theorem 1. Let $\gamma \ge \frac{C_{\min}}{\kappa^2 C_{\max}}\sqrt{\frac{n}{\log p}}\max_{l \ne j}\frac{|\Sigma_{lj}|}{\Sigma_{jj}}$. If $\boldsymbol{A}$ obeys properties **D1**, **D2** and*

$n \geq \frac{8C_{\max}^2 \kappa^4}{C_{\min}^2 (1-c)^2} \log p$ *for some* $c \in (0,1)$, *then*

$$P\left(\frac{\rho}{1+\rho} \leq (2\sqrt{2}+\gamma)\frac{\kappa^2}{c}\frac{C_{\max}}{C_{\min}}\sqrt{\frac{\log p}{n}}\right) \geq 1 - \frac{3}{2p^2}. \tag{8}$$

*Furthermore, if* $c \in \left(\frac{2\sqrt{2}+\gamma}{4\sqrt{2}+\gamma}, 1\right)$ *and* $\mu := (2\sqrt{2}+\gamma)\frac{\kappa^2}{c}\frac{C_{\max}}{C_{\min}}\sqrt{\frac{\log p}{n}}$, *then with high probability,* (6) *is feasible and the optimal debiasing matrix* $\boldsymbol{W}$ *in* (6) *is given by* (7).

The proof of Theorem 2 is given in Appendix A.2.

**Remarks:**

1. If $\boldsymbol{\Sigma}$ is a diagonal matrix (i.e., entries of the sensing matrix $\boldsymbol{A}$ are uncorrelated), then we can choose $\gamma = 0$ and the upper bound of $\rho/(1+\rho)$ in (8) reduces to $2\sqrt{2}\frac{\kappa^2}{c}\frac{C_{\max}}{C_{\min}}\sqrt{\frac{\log p}{n}}$ for any constant $c \in (1/2, 1)$. If $\boldsymbol{\Sigma}$ is not a diagonal matrix, the parameter $\gamma$ represents a degree of dependence between the elements of the rows of the matrix $\boldsymbol{A}$.

2. The condition $c \in \left(\frac{2\sqrt{2}+\gamma}{4\sqrt{2}+\gamma}, 1\right)$ in Theorem 2 ensures that when $n \geq \frac{8C_{\max}^2 \kappa^4}{C_{\min}^2 (1-c)^2} \log p$ and $\mu := (2\sqrt{2}+\gamma)\frac{\kappa^2}{c}\frac{C_{\max}}{C_{\min}}\sqrt{\frac{\log p}{n}}$, then we have $\mu < 1$.

3. In practice, one tends to choose a small value of $\mu$ for debiasing the LASSO estimator. Given $n$, $p$ and $\boldsymbol{\Sigma}$, the exact distribution of $\frac{\rho}{1+\rho}$ can also be estimated with high precision through simulations. One may also choose $\mu$ to be slightly larger than the maximum support of the distribution of $\frac{\rho}{1+\rho}$. This produces a simple and elegant way to choose $\mu$ in practice. Given a fixed $\boldsymbol{A}$, it is easy to compute $\rho$, by definition of $\rho$ in Theorem 1. So we can simply set $\mu := \frac{\rho}{1+\rho}$ since smaller (but feasible) values of $\mu$ are desirable in (6). In Sec.5.4, we observe that this empirical choice of $\mu$ is smaller than the choice of $\mu := (2\sqrt{2}+\gamma)\frac{\kappa^2}{c}\frac{C_{\max}}{C_{\min}}\sqrt{\frac{\log p}{n}}$.

## 4 Relation to Recent Developments in LASSO Debiasing

The LASSO debiasing literature has seen many recent developments in the statistics as well as the AI/ML communities. For instance, the theory of the debiased LASSO has been extended to handle generalized linear models in Vazquez & Nan (2025); Xia et al. (2023). Applications of the debiased LASSO (extended to handle the total variation image prior) to compressive reconstruction of magnetic resonance images have been recently explored in Hoppe et al. (2024b). Bootstrap approaches to further diminish the bias value in low-sample regimes have been explored in Li (2020). Along similar lines, a learning based technique for further reduction of the bias in small-sample regimes has been recently explored in Hoppe et al. (2024a), where it is also shown how to incorporate debiasing for data-driven approaches such as unrolled neural networks. Debiasing techniques have also been extended to handle sparse quantile regression in Yan et al. (2023). Note that all these techniques in principle require computation of either the approximate covariance matrix $\boldsymbol{M}$ or the debiasing matrix $\boldsymbol{W}$, after which various other steps in their approach are carried out. Since our work in this paper proposes a technique to speedily compute $\boldsymbol{W} = \boldsymbol{A}\boldsymbol{M}^\top$, it is clear that it can be readily incorporated to speed up the key step of estimation of $\boldsymbol{M}$ or $\boldsymbol{W}$ in each of these aforementioned approaches.

## 5 Empirical Results

The MATLAB codes for the simulations in this section and also those from Sec. 6 are available at `https://github.com/Shuvayan21/Fast-Debiasing`.

### 5.1 Difference between the exact closed form solution $W_e$ and the solution of the optimization problem in (6) given by $W_o$ for varying choices of $\mu$

**Aim:** In Theorem 1, we show that if $\frac{\rho}{1+\rho} \leq \mu < 1$, then the exact closed form solution of (7) represented by $W_e$ is the same as the solution of the optimization problem given in (6) represented by $W_o$. In this subsection, we investigate the difference between $W_o$ and $W_e$ for $\mu < \frac{\rho}{1+\rho}$ as well as in the range $\frac{\rho}{1+\rho} \leq \mu < 1$. We report the difference between $W_e$ and $W_o$ in terms of the *Relative Error* given by $\left(\frac{\|W_o - W_e\|_F}{\|W_e\|_F}\right)$ for $\mu = 0.2, 0.21, 0.22, \ldots, 0.60$.

**Sensing matrix properties:** For this experiment, we fixed $n = 80, p = 100$. We ran this experiment for two different $n \times p$ sensing matrices $A$ with elements drawn from: (1) i.i.d. Gaussian and, (2) i.i.d. Rademacher. In Figure 1, we plot $\mu$ vs $\left(\frac{\|W_o - W_e\|_F}{\|W_e\|_F}\right)$ for both of these matrices on a log scale. The exact value of $\frac{\rho}{1+\rho}$ is given by a black vertical line in each case.

**Observation:** We see that for both the plots in Figure 1, the relative error decreases with increase in $\mu$ for $\mu < \frac{\rho}{1+\rho}$. For $\mu \geq \frac{\rho}{1+\rho}$, the relative error is very small with fluctuations primarily due to the solver tolerances in `lsqlin` when computing $W_o$. Furthermore, the decrease in relative error is sharp after the value of $\mu$ crosses $\frac{\rho}{1+\rho}$.

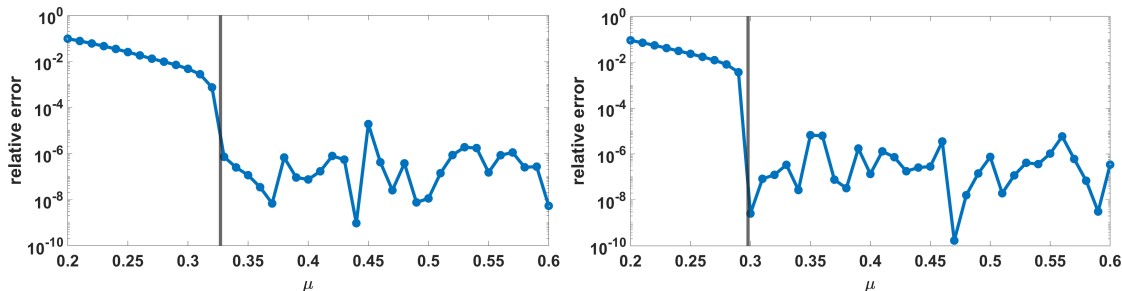

Figure 1: Line plot of $\mu$ vs relative error $\left(\frac{\|W_o - W_e\|_F}{\|W_e\|_F}\right)$ (in log scale) for two $80 \times 100$ dimensional sensing matrices: (left) i.i.d. Gaussian and (right) i.i.d. Rademacher. The exact value of $\frac{\rho}{1+\rho}$ is given by the black vertical line. The value of $\frac{\rho}{1+\rho}$ is 0.327 for the Gaussian sensing matrix (left) and 0.298 for the Rademacher sensing matrix (right). Here, $W_o$ is the solution of the optimization problem in (6) and $W_e$ is computed as in (7).

### 5.2 Comparison of debiasing performance using the exact solution $W_e$ and the choice $M := d\Sigma^{-1}$

In this subsection, we compare the sensitivity, specificity of the debiased estimate $\hat{\beta}_{W_e}$ obtained from (5) using the exact closed-form solution $W_e$, and the debiased estimate $\hat{\beta}_{d\Sigma^{-1}}$ given in Equation (17) of Javanmard & Montanari (2014b) with the debiasing matrix $M := d\Sigma^{-1}$ (note that $M$ is an approximate inverse of the empirical covariance matrix $\hat{\Sigma}$ – see Sec. 2) where $d := (1 - \|\hat{\beta}_{\lambda_1}\|_0/p)^{-1}$. We further compare the ratios of the empirical total variance (ETV) and asymptotic total variance (ATV) of $\hat{\beta}_{W_e}$ and $\hat{\beta}_{d\Sigma^{-1}}$. The ATVs of the $j$th element of $\hat{\beta}_{W_e}$ and $\hat{\beta}_{d\Sigma^{-1}}$ are respectively given by $\frac{1}{n}\sum_{j=1}^{p} w_{.j}^\top w_{.j}$ and $\frac{d^2}{n}\sum_{j=1}^{p}[\Sigma^{-1}A^\top A\Sigma^{-1\top}]_{jj}$. The empirical total variance (ETV) of the debiased LASSO estimators is obtained using 100 simulation runs over different instances of $\eta$ with varying $n \in \{250, 350, 500\}$, $f_\sigma = 0.01$, $p = 500$, $s = 5$ where the signal $\beta^*$ was generated in the same manner as described in the beginning of this section. In these simulations, the rows of $A$ are generated as $p$-dimensional i.i.d. random vectors from $\mathcal{N}_p(\mathbf{0}, \Sigma)$ for three different choices of $\Sigma$ given as follows:

1. **Diagonal Matrix:** $\Sigma_1 = \sigma^2 I_p$ with choice $\sigma^2 = 1$.

2. **Banded Equicorrelated Matrix: $\boldsymbol{\Sigma_2}$** with $(i,j)^{\text{th}}$ entry as follows.

$$\Sigma_{2_{ij}} = \begin{cases} \sigma^2, & \text{if } i = j \in [p], \\ \sigma^2\zeta, & \text{if } |i - j| \leq b, \ i \neq j \in [p] \ , \\ 0, & \text{otherwise} \end{cases} \text{ with choices } \zeta = 0.1, \ b = 5 \text{ and } \sigma^2 = 1.$$

3. **Equicorrelated Matrix: $\boldsymbol{\Sigma_3}$** $:= \sigma^2\big[(1-\zeta)\boldsymbol{I}_p + \zeta\boldsymbol{1}_p\boldsymbol{1}_p^\top\big]$ with choices $\zeta = 0.1$ and $\sigma^2 = 1$. Here, $\boldsymbol{1}_p$ denotes the $p$-dimensional vector of all ones.

The diagonal covariance matrices are widely used in compressed sensing Candès et al. (2006). The chosen banded equicorrelated covariance matrix ($\boldsymbol{\Sigma_2}$) is a special case of a symmetric circulant matrix, which has been explored by Javanmard & Montanari (2014a) in the context of debiasing the Lasso estimator. Further, the equicorrelated matrix $\boldsymbol{\Sigma_3}$ has a motivation in compressed sensing as well, to express cross-talk—given by the term $\zeta\boldsymbol{1}\boldsymbol{1}^\top$—between different elements of a sensor array. In single-pixel cameras (a common architecture in compressed sensing) Duarte et al. (2008), the term the term $\zeta\boldsymbol{1}_p\boldsymbol{1}_p^\top$ models global illumination changes (similar to a background interference) which bring in weak correlation, so that the $j$th row of the sensing matrix can be effectively expressed by $\tilde{\boldsymbol{a}}^j = \sqrt{1-\zeta}\boldsymbol{a}^j + \sqrt{\zeta}\boldsymbol{1}_p^\top$.

In Tables 1, 2 and 3, we present results for each of these covariance designs comparing Sensitivity and Specificity, for both debiased estimates (using $\boldsymbol{W}_e$ and $d\boldsymbol{\Sigma}^{-1}$). We also present the ratios of the ATV and ETV for these estimates.

| $n$ | $\text{Sens}(\hat{\boldsymbol{\beta}}_{\boldsymbol{W}_e})$ | $\text{Spec}(\hat{\boldsymbol{\beta}}_{\boldsymbol{W}_e})$ | $\text{Sens}(\hat{\boldsymbol{\beta}}_{d\boldsymbol{\Sigma}^{-1}})$ | $\text{Spec}(\hat{\boldsymbol{\beta}}_{d\boldsymbol{\Sigma}^{-1}})$ | $\frac{ATV(\hat{\boldsymbol{\beta}}_{\boldsymbol{W}_e})}{ATV(\hat{\boldsymbol{\beta}}_{d\boldsymbol{\Sigma}^{-1}})}$ | $\frac{ETV(\hat{\boldsymbol{\beta}}_{\boldsymbol{W}_e})}{ETV(\hat{\boldsymbol{\beta}}_{d\boldsymbol{\Sigma}^{-1}})}$ |
|---|---|---|---|---|---|---|
| 250 | 0.7145 | 0.8972 | 0.7209 | 0.8653 | 0.2819 | 0.3863 |
| 350 | 0.8554 | 0.9719 | 0.8126 | 0.9233 | 0.3882 | 0.5182 |
| 500 | 0.9985 | 0.9992 | 0.9486 | 0.9492 | 0.4699 | 0.6075 |

Table 1: **Diagonal Matrix $\boldsymbol{\Sigma_1}$:** (see Sec. 5.4) Comparison of sensitivity, specificity, and ATV, ETV ratios for the debiased estimates $\hat{\boldsymbol{\beta}}_{\boldsymbol{W}_e}$ and $\hat{\boldsymbol{\beta}}_{d\boldsymbol{\Sigma}^{-1}}$ across different sample sizes $n \in [200 : 50 : 500]$ for an uncorrelated Gaussian design matrix. The fixed parameters are $p = 500, f_\sigma = 0.01, s = 5, r = 4$.

| $n$ | $\text{Sens}(\hat{\boldsymbol{\beta}}_{\boldsymbol{W}_e})$ | $\text{Spec}(\hat{\boldsymbol{\beta}}_{\boldsymbol{W}_e})$ | $\text{Sens}(\hat{\boldsymbol{\beta}}_{d\boldsymbol{\Sigma}^{-1}})$ | $\text{Spec}(\hat{\boldsymbol{\beta}}_{d\boldsymbol{\Sigma}^{-1}})$ | $\frac{ATV(\hat{\boldsymbol{\beta}}_{\boldsymbol{W}_e})}{ATV(\hat{\boldsymbol{\beta}}_{d\boldsymbol{\Sigma}^{-1}})}$ | $\frac{ETV(\hat{\boldsymbol{\beta}}_{\boldsymbol{W}_e})}{ETV(\hat{\boldsymbol{\beta}}_{d\boldsymbol{\Sigma}^{-1}})}$ |
|---|---|---|---|---|---|---|
| 250 | 0.7392 | 0.8871 | 0.6975 | 0.8387 | 0.2573 | 0.3982 |
| 350 | 0.8833 | 0.9562 | 0.8136 | 0.8865 | 0.3142 | 0.5961 |
| 500 | 0.9715 | 0.9854 | 0.9006 | 0.9216 | 0.3924 | 0.7269 |

Table 2: **Banded Equicorrelated Matrix $\boldsymbol{\Sigma_2}$:** (see Sec. 5.4) Comparison of sensitivity, specificity, and ATV, ETV ratios for the debiased estimates $\hat{\boldsymbol{\beta}}_{\boldsymbol{W}_e}$ and $\hat{\boldsymbol{\beta}}_{d\boldsymbol{\Sigma}^{-1}}$ across different sample sizes $n = [200 : 50 : 500]$ for correlated Gaussian design given as a bandwidth-3 matrix with $\Sigma_{ij} = \sigma^2 \cdot 0.1, \quad |i - j| \leq 3$, and zero otherwise. The fixed parameters are $p = 500, f_\sigma = 0.01, s = 5, r = 4$.

| $n$ | $\text{Sens}(\hat{\boldsymbol{\beta}}_{\boldsymbol{W}_e})$ | $\text{Spec}(\hat{\boldsymbol{\beta}}_{\boldsymbol{W}_e})$ | $\text{Sens}(\hat{\boldsymbol{\beta}}_{d\boldsymbol{\Sigma}^{-1}})$ | $\text{Spec}(\hat{\boldsymbol{\beta}}_{d\boldsymbol{\Sigma}^{-1}})$ | $\frac{ATV(\hat{\boldsymbol{\beta}}_{\boldsymbol{W}_e})}{ATV(\hat{\boldsymbol{\beta}}_{d\boldsymbol{\Sigma}^{-1}})}$ | $\frac{ETV(\hat{\boldsymbol{\beta}}_{\boldsymbol{W}_e})}{ETV(\hat{\boldsymbol{\beta}}_{d\boldsymbol{\Sigma}^{-1}})}$ |
|---|---|---|---|---|---|---|
| 250 | 0.7275 | 0.8655 | 0.6855 | 0.8152 | 0.2724 | 0.3892 |
| 350 | 0.8112 | 0.9216 | 0.7908 | 0.8872 | 0.3433 | 0.4844 |
| 500 | 0.9466 | 0.9573 | 0.9212 | 0.9319 | 0.4147 | 0.5795 |

Table 3: **Equicorrelated Matrix $\boldsymbol{\Sigma_3}$:** (see Sec. 5.4) Comparison of sensitivity, specificity, and ATV, ETV ratios for the debiased estimates $\hat{\boldsymbol{\beta}}_{\boldsymbol{W}_e}$ and $\hat{\boldsymbol{\beta}}_{d\boldsymbol{\Sigma}^{-1}}$ across different sample sizes $n = [200 : 50 : 500]$ for correlated Gaussian design given as a bandwidth-3 matrix with $\Sigma_{ij} = \sigma^2 \cdot 0.1, \quad |i - j| \leq 3$, and zero otherwise. The fixed parameters are $p = 500, f_\sigma = 0.01, s = 5, r = 4$.

From Tables 1, 2 and 3, it is evident that the debiased estimator $\hat{\boldsymbol{\beta}}_{\boldsymbol{W}_e}$ consistently outperforms $\hat{\boldsymbol{\beta}}_{d\boldsymbol{\Sigma}^{-1}}$ in terms of sensitivity and specificity, with the advantage being more pronounced for smaller sample sizes and gradually diminishing as $n$ increases. This performance benefit is not surprising because in our approach, the matrix $\boldsymbol{W}$ is specifically *designed* to produce a debiased estimator of *minimum variance*, unlike the choice of $\boldsymbol{M} := d\boldsymbol{\Sigma}^{-1}$ which only provides debiasing. Furthermore, the debiasing properties of $\boldsymbol{M} := d\boldsymbol{\Sigma}^{-1}$ have only been established for Gaussian uncorrelated designs in Javanmard & Montanari (2014b), whereas our approach is applicable to a much wider range of matrices. Moreover, our approach does not require knowledge of $\boldsymbol{\Sigma}$, which may not be available and is hard to estimate even for uncorrelated designs because $n < p$. Lastly, our approach does not rely on the $\ell_0$ norm of the LASSO estimate.

The introduction of correlation in the design matrix leads to an overall reduction in both sensitivity and specificity at lower $n$, but this gap narrows down with larger $n$. Furthermore, the variance ratios remain below unity across all settings, indicating that $\hat{\boldsymbol{\beta}}_{\boldsymbol{W}_e}$ achieves lower empirical and asymptotic variances, with the ratios increasing steadily in $n$, reflecting greater stability. Overall, $\hat{\boldsymbol{\beta}}_{\boldsymbol{W}_e}$ demonstrates superior efficiency and robustness to correlation compared to $\hat{\boldsymbol{\beta}}_{d\boldsymbol{\Sigma}^{-1}}$.

### 5.3 Validity of the exact solution

**Aim:** The debiased LASSO can be used to determine the support of the unknown vector $\boldsymbol{\beta}^*$ by using statistical hypothesis tests derived using LASSO debiasing theory. We aim to estimate the support using $p$ hypothesis tests (one per element of $\boldsymbol{\beta}^*$) based on the debiased LASSO estimates using the weights matrix $\boldsymbol{W}$ obtained from the optimization problem in (6) (denoted by $\boldsymbol{W_o}$), and that obtained from the closed-form expression (7) (denoted by $\boldsymbol{W_e}$), for varying number of measurements $n$. The aim is to also compare these support set estimates with the ground truth support set, and report sensitivity and specificity values (defined below). We will further show the difference in the run-time for both methods.

**Signal Generation:** For our simulations, we chose our design matrix $\boldsymbol{A}$ to have elements drawn independently from the standard Gaussian distribution. We synthetically generated signals (i.e., $\boldsymbol{\beta}^*$) with $p = 500$ elements in each. The non-zero values of $\boldsymbol{\beta}^*$ were drawn i.i.d. from $U(50, 1000)$ and placed at randomly chosen indices. We set $s := \|\boldsymbol{\beta}^*\|_0 = 10$ and the noise standard deviation $\sigma := 0.05 \sum_{i=1}^{n} |\boldsymbol{a_{i.}}\boldsymbol{\beta}^*|/n$. We varied $n \in \{200, 250, 300, 350, 400, 450, 500\}$. We chose $\mu = \rho/(\rho + 1)$ where $\rho$ was computed exactly given the sensing matrix $\boldsymbol{A}$.

**Sensitivity and Specificity Computation:** Let us denote the debiased LASSO estimates obtained using a matrix $\boldsymbol{W}$ by $\hat{\beta}_{d,W}$. We know that asymptotically $\hat{\beta}_{d,W}(j) \sim \mathcal{N}(\beta_j^*, \sigma^2 \boldsymbol{w}_{.j}^\top \boldsymbol{w}_{.j}/n^2)$ for all $j \in [p]$. Using this result, $\hat{\boldsymbol{\beta}}_{d,W}$ was binarized to create a vector $\hat{\boldsymbol{b}}_W$ in the following way: For all $j \in [p]$, we set $\hat{b}_W(j) := 1$ if the value of $\hat{\beta}_W(j)$ was such that the the hypothesis $\mathsf{H}_{0,j} : \beta_j^* = 0$ was rejected against the alternate $\mathsf{H}_{1,j} : \beta_j^* \neq 0$ at 5% level of significance. $\hat{b}_W(j)$ was set to 0 otherwise. Note that for the purpose of our simulation, we either have $\boldsymbol{W} = \boldsymbol{W_o}$ or $\boldsymbol{W} = \boldsymbol{W_e}$. The binary vectors corresponding to these choices of $\boldsymbol{W}$ are respectively denoted by $\hat{\boldsymbol{b}}_{\boldsymbol{W_o}}$ and $\hat{\boldsymbol{b}}_{\boldsymbol{W_e}}$.

A ground truth binary vector $\boldsymbol{b}^*$ was created such that $b_j^* := 1$ at all locations $j$ where $\beta_j^* \neq 0$ and $b_j^* := 0$ otherwise. Sensitivity and specificity values were computed by comparing corresponding entries of $\boldsymbol{b}^*$ to those in $\hat{\boldsymbol{b}}_{\boldsymbol{W_o}}$ and $\hat{\boldsymbol{b}}_{\boldsymbol{W_e}}$. Considering the matrix $\boldsymbol{W}$, we declared an element to be a *true defective* if $b_j^* = 1$ and $\hat{b}_W(j) = 1$, and a *false defective* if $b_j^* = 0$ but $\hat{b}_W(j) \neq 0$. We declare it to be a *false non-defective* if $b_j^* = 0$ but $\hat{b}_W(j) \neq 0$, and a *true non-defective* if $\beta_j^* = 0$ and $\hat{b}_W(j) = 0$. The **sensitivity** for $\boldsymbol{\beta}^*$ is defined as (# true defectives)/(# true defectives + # false non-defectives) and **specificity** for $\boldsymbol{\beta}^*$ is defined as (# true non-defectives)/(# true non-defectives + # false defectives).

**Results:** For obtaining $\boldsymbol{W_o}$, the optimization routine was executed using the `lsqlin` package in MATLAB. The sensitivity and specificity were averaged over 25 runs with independent noise instances.

In Table 4, we can see that the sensitivity as well as the specificity of the hypothesis tests for $\boldsymbol{W_o}$ and $\boldsymbol{W_e}$ are equal. We further report the relative difference between $\boldsymbol{W_o}$ and $\boldsymbol{W_e}$ in the Frobenius norm. We

| | sensitivity | | specificity | | time (in s) | | |
|---|---|---|---|---|---|---|---|
| $n$ | $\boldsymbol{W_o}$ | $\boldsymbol{W_e}$ | $\boldsymbol{W_o}$ | $\boldsymbol{W_e}$ | $\boldsymbol{W_o}$ | $\boldsymbol{W_e}$ | $\frac{\|\boldsymbol{W_o}-\boldsymbol{W_e}\|_F}{\|\boldsymbol{W_e}\|_F}$ |
| 200 | 0.6742 | 0.6742 | 0.8592 | 0.8592 | $3.88 \times 10^2$ | $1.11 \times 10^{-3}$ | $6.68 \times 10^{-10}$ |
| 250 | 0.7229 | 0.7229 | 0.9063 | 0.9063 | $5.22 \times 10^2$ | $1.72 \times 10^{-3}$ | $2.31 \times 10^{-8}$ |
| 300 | 0.8071 | 0.8071 | 0.9427 | 0.9427 | $3.29 \times 10^2$ | $2.25 \times 10^{-3}$ | $2.73 \times 10^{-7}$ |
| 350 | 0.8554 | 0.8554 | 0.9719 | 0.9719 | $4.77 \times 10^2$ | $3.88 \times 10^{-3}$ | $2.56 \times 10^{-7}$ |
| 400 | 0.9275 | 0.9275 | 0.9855 | 0.9855 | $5.59 \times 10^2$ | $7.82 \times 10^{-3}$ | $4.76 \times 10^{-7}$ |
| 450 | 0.9781 | 0.9781 | 0.9909 | 0.9909 | $7.15 \times 10^2$ | $4.27 \times 10^{-2}$ | $5.29 \times 10^{-7}$ |
| 500 | 0.9985 | 0.9985 | 0.9992 | 0.9992 | $8.03 \times 10^2$ | $7.56 \times 10^{-2}$ | $8.22 \times 10^{-7}$ |

Table 4: Sensitivity and Specificity of hypothesis test using debiased estimates obtain from $\boldsymbol{W_o}$ (optimization method) and $\boldsymbol{W_e}$ (closed-form expression from (7)) with its corresponding runtime in seconds for varying number of measurements. The fixed parameters are $p = 500, s = 10, \sigma := 0.05 \sum_{i=1}^n |\boldsymbol{a_{i.}\beta^*}|/n$. We set $\mu = \rho/(\rho + 1)$ where $\rho$ is computed exactly for the chosen sensing matrix $\boldsymbol{A}$.

| | sensitivity | | specificity | | time (in s) | | |
|---|---|---|---|---|---|---|---|
| $\mu$ | $\boldsymbol{W_o}$ | $\boldsymbol{W_e}$ | $\boldsymbol{W_o}$ | $\boldsymbol{W_e}$ | $\boldsymbol{W_o}$ | $\boldsymbol{W_e}$ | $\frac{\|\boldsymbol{W_o}-\boldsymbol{W_e}\|_F}{\|\boldsymbol{W_e}\|_F}$ |
| 0.2 | 0.9586 | 0.9544 | 0.9942 | 0.9901 | $8.44 \times 10^2$ | $7.76 \times 10^{-3}$ | $2.24 \times 10^{-1}$ |
| 0.25 | 0.9531 | 0.9502 | 0.9872 | 0.9855 | $6.91 \times 10^2$ | $8.72 \times 10^{-3}$ | $7.62 \times 10^{-2}$ |
| 0.3 | 0.9475 | 0.9475 | 0.9921 | 0.9921 | $5.59 \times 10^2$ | $8.12 \times 10^{-3}$ | $3.39 \times 10^{-7}$ |
| 0.35 | 0.9354 | 0.9354 | 0.9891 | 0.9891 | $5.42 \times 10^2$ | $7.83 \times 10^{-3}$ | $6.312 \times 10^{-7}$ |
| 0.4 | 0.9275 | 0.9275 | 0.9855 | 0.9855 | $5.77 \times 10^2$ | $7.56 \times 10^{-3}$ | $2.08 \times 10^{-8}$ |
| 0.45 | 0.9102 | 0.9102 | 0.9792 | 0.9792 | $5.98 \times 10^2$ | $7.49 \times 10^{-3}$ | $4.55 \times 10^{-7}$ |

Table 5: Sensitivity and Specificity of hypothesis tests using debiased estimates obtained using $\boldsymbol{W_o}$ (optimization method) and $\boldsymbol{W_e}$ (closed-form expression from (7)) for varying choice of $\mu$. The corresponding run-times for estimating $\boldsymbol{W_o}$ and $\boldsymbol{W_e}$ from a Rademacher sensing matrix $\boldsymbol{A}$, are also shown. The fixed parameters are $p = 500, n = 400, s = 10, \sigma := 0.05 \sum_{i=1}^n |\boldsymbol{a_{i.}\beta^*}|/n$. The exact value of $\frac{\rho}{1+\rho} = 0.298$ where $\rho$ is computed exactly for the chosen sensing matrix $\boldsymbol{A}$.

can clearly see that the difference is negligible, which is consistent with Theorem 1. Furthermore, we see that using the closed-form expression in (7) saves significantly on time (by a factor of at least $10^4$). While the computational efficiency of the iterative approach can be improved by developing a specialized solver for problems of the form (6), no iterative method is expected to outperform directly computing the simple closed-form expression (7).

In Table 5, we observe that both debiasing matrices $\boldsymbol{W_o}$ and $\boldsymbol{W_e}$ exhibit almost identical sensitivity and specificity for the hypothesis tests across a wide range of $\mu$ values greater than or equal to 0.2 to 1. This range was chosen because we observed that for the choices of $\mu < 0.2$, the optimization problem (4) was often not feasible. For $\mu > \rho/(1+\rho) = 0.298$, the sensitivity and specificity of the debiasing methods with $W_o$ and $W_e$ was the same (up to numerical tolerances in the optimizer) which is consistent with our theory. For $\mu > 0.45$, the sensitivity and specificity of LASSO debiasing with both $\boldsymbol{W_o}$ and $\boldsymbol{W_e}$ was below 0.9 (not shown in the table), but it remained identical for both methods. For $0.2 < \mu < 0.298$, the sensitivity and specificity values with the two methods were similar even though not identical. In all cases, however, the major distinction between the two methods was computational time, as computing $\boldsymbol{W_o}$ took more than 550 seconds whereas $\boldsymbol{W_e}$ was obtained in a few milliseconds. Given this dramatic speed-up and the similar statistical performance, the closed-form $\boldsymbol{W_e}$ offers a highly practical and efficient alternative to the optimization-based solution $\boldsymbol{W_o}$.

### 5.4 Empirical Distribution of $\rho/(1+\rho)$

In this subsection, we will show that the support of the distribution of $\frac{\rho}{1+\rho}$ is smaller than the choice of $\mu := (2\sqrt{2} + \gamma)\frac{\kappa^2}{c}\frac{C_{\max}}{C_{\min}}\sqrt{\frac{\log p}{n}}$ given by Theorem 2 for the different chosen covariance matrices $\boldsymbol{\Sigma_1}, \boldsymbol{\Sigma_2}, \boldsymbol{\Sigma_3}$ defined in Sec. 5.2. We chose $p = 500$ and $n \in \{250, 350, 500\}$. For each configuration, we generated 1000 independent $n \times p$ matrices $\boldsymbol{A}$, with rows sampled i.i.d. from $\mathcal{N}_p(0, \boldsymbol{\Sigma})$.

For each realization of $\boldsymbol{A}$, we computed $\rho(\boldsymbol{A}) = \max_{i \neq j} \frac{|\boldsymbol{a}_{\cdot i}^\top \boldsymbol{a}_{\cdot j}|}{\|\boldsymbol{a}_{\cdot j}\|_2^2}$. The normalized histograms of $\rho/(1+\rho)$ based on 1000 simulation runs are shown in Figure 2. The top, middle and bottom rows of Figure 2 respectively correspond to the covariance matrix $\boldsymbol{\Sigma_1}$, $\boldsymbol{\Sigma_2}$ and $\boldsymbol{\Sigma_3}$, whereas the left, center and right column respectively correspond to $n = 250, 350$ and $500$. Each plot is overlaid with a red vertical line showing the bound $\mu := (2\sqrt{2} + \gamma)\frac{\kappa^2}{c}\frac{C_{\max}}{C_{\min}}\sqrt{\frac{\log p}{n}}$ as given in Theorem 2. Ideally, we would like to choose $\gamma$ as small as possible and $c$ to be as large as possible. Therefore in our experiments, we chose $\gamma = \frac{C_{\min}}{\kappa^2 C_{\max}}\sqrt{\frac{n}{\log p}}\max_{l \neq j}\frac{|\Sigma_{lj}|}{\Sigma_{jj}}$. Furthermore, under the assumption $n \geq \frac{8C_{\max}^2 \kappa^4}{C_{\min}^2(1-c)^2}\log p$, we chose $c = 1 - 2\sqrt{2}\frac{\kappa^2}{c}\frac{C_{\max}}{C_{\min}}\sqrt{\frac{\log p}{n}}$.

In Figure 2, we observe that the span of the normalized histograms shrinks and moves towards the origin for all the chosen covariance matrices. This indicates that $\frac{\rho}{1+\rho}$ tends to zero as the sample size increases. Figure 2 also shows that the upper bound on $\frac{\rho}{1+\rho}$ obtained from Theorem 2 is conservative (in terms of constant factors) for smaller sample sizes. We also observe that the probability density of $\frac{\rho}{1+\rho}$ depends on the dependent structure of $\boldsymbol{\Sigma}$. Therefore, given the values of $n$, $p$ and $\boldsymbol{\Sigma}$, one may choose $\mu$ which is slightly larger than the maximum value of the support of the distribution of $\frac{\rho}{1+\rho}$ in practice, which can be obtained using simulation before performing the debiasing.

## 6 Experiments on Compressive Image/Video Reconstruction

We further validated the use of our fast debiasing approach for image reconstruction from noisy compressive measurements given three different compressed sensing architectures (that is, realistic models for $\boldsymbol{A}$): the Rice Single Pixel Camera for compressive imaging Duarte et al. (2008), the coded exposure snapshot camera for compressive video acquisition Liu et al. (2013) and the coded aperture snapshot spectral imager (CASSI) for hyperspectral image acquisition Kittle et al. (2010). There already exist a plethora of compressive image reconstruction techniques, both classical Foucart & Rauhut (2013) and deep learning based Kulkarni et al. (2016). Here, our aim is to provide proof of concept that fast debiasing is applicable to realistic sensing matrices; our aim here is not to beat the state of the art. However, compared to the existing techniques, our

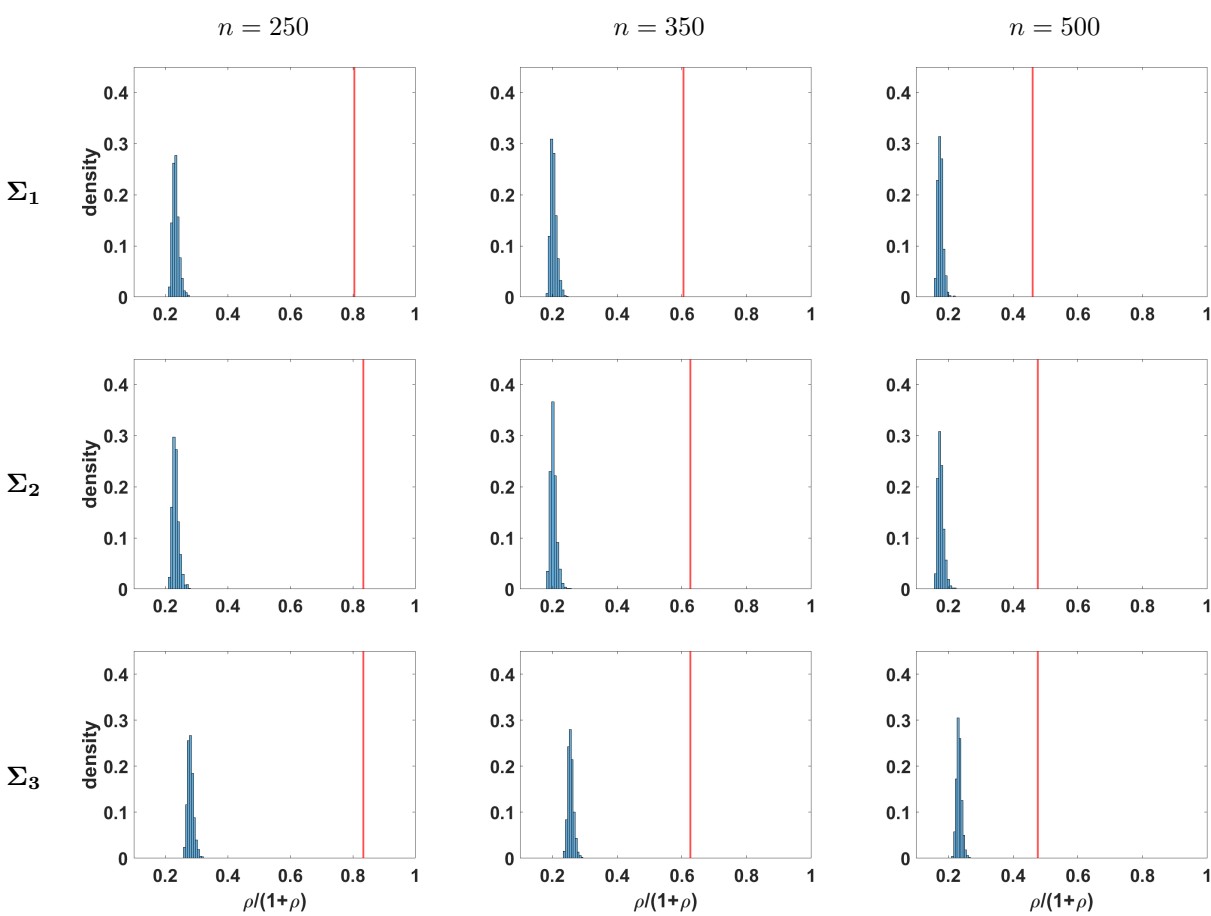

Figure 2: Histograms of $\rho/(1+\rho)$ based on 1000 simulation runs for $p = 500$ and $n \in \{250, 350, 500\}$. Rows correspond to different covariance structures (diagonal, banded equicorrelated and equicorrelated), while columns correspond to sample size $n$. The red lines indicate the theoretical benchmark of $\mu := (2\sqrt{2} + \gamma)\frac{\kappa^2}{c}\frac{C_{\max}}{C_{\min}}\sqrt{\frac{\log p}{n}}$ by choosing $c = 1 - 2\sqrt{2}\frac{\kappa^2}{c}\frac{C_{\max}}{C_{\min}}\sqrt{\frac{\log p}{n}}$ and $\gamma = \frac{C_{\min}}{\kappa^2 C_{\max}}\sqrt{\frac{n}{\log p}}\max_{l \neq j}\frac{|\Sigma_{lj}|}{\Sigma_{jj}}$ for all designs.

presented approach is unique in its ability to provide *quantification of the uncertainty* in the reconstructed pixel values, an aspect which the aforementioned techniques do not cover.

Consider a noisy measurement vector $\boldsymbol{y} \in \mathbb{R}^n$ of the form $\boldsymbol{y} = \boldsymbol{\Phi}\boldsymbol{f} + \boldsymbol{\eta} = \boldsymbol{\Phi}\boldsymbol{\Psi}\boldsymbol{\theta} + \boldsymbol{\eta}$, where $\boldsymbol{\eta}$ is an additive noise vector whose elements are drawn independently from $\mathcal{N}(0, \sigma^2)$, $\boldsymbol{\Phi} \in \mathbb{R}^{n \times p}$ is a sensing matrix of i.i.d. sub-Gaussian distributed entries, and $\boldsymbol{\Psi} \in \mathbb{R}^{p \times p}$ is an orthonormal basis in which the image $\boldsymbol{f}$ is sparse or 'weakly sparse' – that is $\boldsymbol{f} = \boldsymbol{\Psi}\boldsymbol{\theta}$, where $\boldsymbol{\theta} \in \mathbb{R}^p$ is a vector of coefficients of which a small number have large magnitude and the majority are either zero (sparse $\boldsymbol{\theta}$) or close to zero (weakly sparse $\boldsymbol{\theta}$). The aim is to reconstruct $\boldsymbol{\theta}$, and thus $\boldsymbol{f}$, from $\boldsymbol{y}, \boldsymbol{\Phi}, \boldsymbol{\Psi}$ in the compressive regime where $n \ll p$. For this estimation task, the LASSO is used: $\hat{\boldsymbol{\theta}} := \arg\min_{\boldsymbol{\theta}} \|\boldsymbol{y} - \boldsymbol{\Phi}\boldsymbol{\Psi}\boldsymbol{\theta}\|_2^2 + \lambda\|\boldsymbol{\theta}\|_1$. This is followed by a debiasing step of the form $\hat{\boldsymbol{\theta}}_{\boldsymbol{d}} = \hat{\boldsymbol{\theta}} + \frac{1}{n}\widetilde{\boldsymbol{W}}^\top(\boldsymbol{y} - \boldsymbol{\Phi}\boldsymbol{\Psi}\hat{\boldsymbol{\theta}})$ where the debiasing matrix $\widetilde{\boldsymbol{W}} = (1 - \mu)\boldsymbol{\Phi}\boldsymbol{\Psi}$ with $\mu$ being exactly as defined in Theorem 1 with $\boldsymbol{A} = \boldsymbol{\Phi}\boldsymbol{\Psi}$. We particularly note that in this case, we first estimate $\boldsymbol{\theta}$ since the image $\boldsymbol{f} \in \mathbb{R}^p$ is not sparse in the canonical basis but is (weakly) sparse in the basis $\boldsymbol{\Psi}$. A typical choice for $\boldsymbol{\Psi}$ is the 2D discrete cosine transform (DCT) basis, though many other choices such as wavelets, shearlets, etc., are also possible. We note that Theorems 1 and 2 are applicable to the matrix $\boldsymbol{A} := \boldsymbol{\Phi}\boldsymbol{\Psi}$ for the following reasons: ($i$) As no column of $\boldsymbol{\Phi}$ equals zero, neither does any column of $\boldsymbol{\Phi}\boldsymbol{\Psi}$ and hence Theorem 1 applies. ($ii$) $\boldsymbol{\Phi}$ has zero-mean sub-Gaussian entries and $\boldsymbol{\Psi}$ has bounded entries. Hence the entries of $\boldsymbol{\Phi}\boldsymbol{\Psi}$ are also

|           | RRMSE  | SSIM  | Cov. Prob. (Edges) | Cov. Prob. (Non-Edges) |
|-----------|--------|-------|--------------------|------------------------|
| **Barbara**   | 0.0325 | 0.932 | 0.72 | 0.93 |
| **Cameraman** | 0.0281 | 0.948 | 0.68 | 0.91 |
| **Moon**      | 0.0204 | 0.971 | 0.79 | 0.95 |
| **Male**      | 0.0357 | 0.918 | 0.66 | 0.92 |

Table 6: Reconstruction quality metrics for grayscale images of size $256 \times 256$ using Debiased LASSO. The metrics reported include RRMSE, SSIM, and average coverage probabilities computed separately on edge and non-edge pixel locations based on the Canny edge detection method.

zero-mean sub-Gaussian and they have the same covariance matrix as the rows of $\mathbf{\Phi}$. Hence if the entries of $\mathbf{\Phi}$ are i.i.d. Gaussian, then the rows of $\mathbf{\Phi\Psi}$ remain independent. Hence Theorem 2 applies. Even though the independence assumption may not strictly hold for the rows of $\mathbf{\Phi\Psi}$ for $\mathbf{\Phi}$ from other distributions (e.g., Rademacher or Bernoulli), we have observed excellent empirical results with debiasing even for such cases.

## 6.1 Image reconstruction for the Rice Single Pixel Camera Model

We simulated noisy compressive measurements from four commonly used images of size $256 \times 256$ via a $\mathbf{\Phi}$ matrix with i.i.d. entries drawn from a Bernoulli distribution with success probability 0.5, which is in tune with the architecture of the celebrated Rice Single Pixel Camera Duarte et al. (2008). In our experiments, we used $n = 20,000$ measurements for $p = 256^2$. The original images, and their reconstructed versions via the LASSO and the debiased LASSO, are shown in Fig. 3. After debiasing, the variance of the $i$th estimated coefficient is given by $\sigma^2 [\widetilde{\mathbf{W}}^\top \widetilde{\mathbf{W}}]_{ii}$. Since $\boldsymbol{f} = \mathbf{\Psi\theta}$, the variance of the $i$th pixel of the estimated image (i.e. $\hat{f}_i$) is given by $\sigma^2 [\mathbf{\Psi}\widetilde{\mathbf{W}}^\top \widetilde{\mathbf{W}}\mathbf{\Psi}^\top]_{ii}$. These variance values form a quantitative measure of the uncertainty inherent in the reconstruction. To illustrate this further, the reconstruction experiment for each image was repeated $K = 50$ times using different realizations of the noise $\boldsymbol{\eta}$. The total number of times $N_i$ (out of $K$) for which $f_i$ (the value of the $i$th pixel of $\boldsymbol{f}$) resided in the interval $\left[\hat{f}_i - z_{1-\alpha/2}\sigma\sqrt{[\mathbf{\Psi}\widetilde{\mathbf{W}}^\top \widetilde{\mathbf{W}}\mathbf{\Psi}^\top]_{ii}}, \hat{f}_i + z_{1-\alpha/2}\sigma\sqrt{[\mathbf{\Psi}\widetilde{\mathbf{W}}^\top \widetilde{\mathbf{W}}\mathbf{\Psi}^\top]_{ii}}\right]$ was recorded. This is the confidence interval of probability $1-\alpha$ for $f_i$, and $z_{1-\alpha/2}$ is the percentile of a standard normal distribution at level $1-\alpha/2$. The values of $N_i/K$—termed coverage probabilities—across all $i \in \{1, 2, ..., p\}$ were recorded and plotted as an image in Fig. 3. As is clearly seen in the fourth column of Fig. 3, these probabilities are very high for pixels in smooth regions and they are the least for pixels lying on edges. This is because the 2D DCT is more efficient in representing smooth regions as compared to discontinuities. The ratios along edges can be improved by using direction-sensitive bases such as shearlets Kutyniok & Labate (2012) or learned overcomplete representations Aharon et al. (2006), but we leave a full investigation of these aspects to future work as they are not central to the main theme of this work.

We report the Relative Root Mean Square Error (RRMSE), defined as $\frac{\|\boldsymbol{\theta}^* - \hat{\boldsymbol{\theta}}\|_2}{\|\boldsymbol{\theta}^*\|_2}$, where $\hat{\boldsymbol{\theta}}$ denotes the estimate of the true parameter $\boldsymbol{\theta}^*$. Along with RRMSE, we present the Structural Similarity Index Measure (SSIM), the average coverage probability over edge pixels, and the average coverage probability over non-edge pixels. These quantitative metrics, summarized in Table 6, correspond to the Debiased LASSO estimates for all four benchmark images. The small RRMSE values and high SSIM scores collectively indicate that the the reconstruction quality is very good. Furthermore, the coverage probabilities display the expected spatial behavior: coverage is considerably higher in smooth, non-edge regions and lower in edge regions, consistent with the qualitative observations in Fig. 3.

**Image reconstruction with $\mathbf{\Phi}$ having equicorrelated entries:** Another set of results was obtained where the i.i.d. Bernoulli model for $\mathbf{\Phi}$ was replaced by the equicorrelated model from Sec. 5.2 to model *cross-talk* between different sensory elements of a compressive device. These results, which are presented in Fig. 4 and Table 7, demonstrate a gentle decrease in reconstruction performance with increase in the cross-talk factor $\zeta$ (i.e., increase in correlation in the elements of the rows of the sensing matrix). Nonetheless, the overall reconstruction quality remains strong across all images when using the Debiased LASSO.

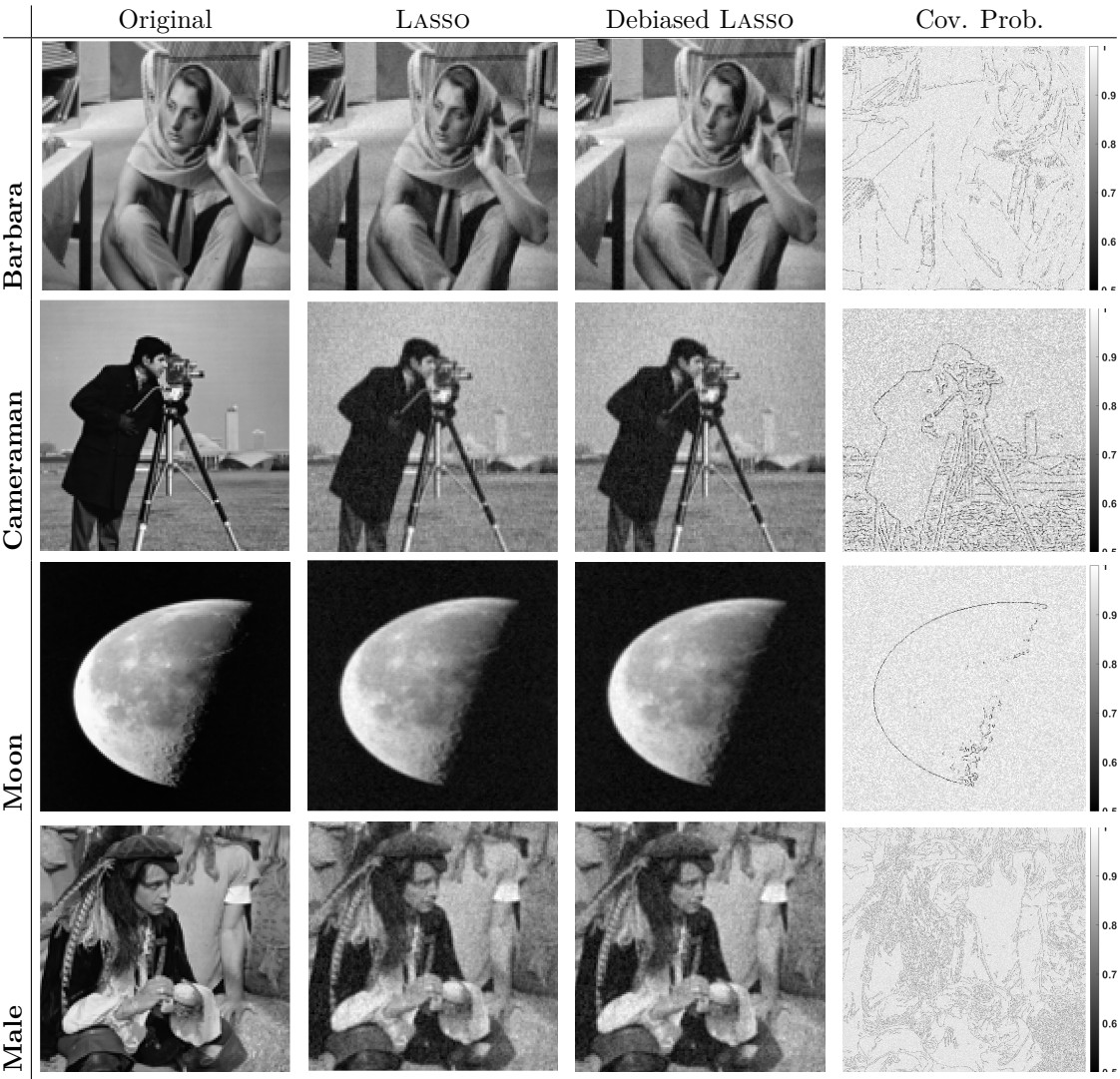

Figure 3: Compressive Image Reconstruction: Reconstruction of grayscale images of dimension $256 \times 256$ (leftmost column) using Lasso (column 2 from left) and Debiased Lasso (column 3 from left) and the empirical coverage probabilities $N_i/K \; \forall i \in [p]$ based on confidence intervals for the debiased Lasso estimates (rightmost column). The sensing matrix contains random Rademacher entries. Here $n = 20000, p = 65,536$ and the additive noise $\sigma = 0.01\times$ the mean absolute value of the noiseless measurements. Debiasing matrix computation using (4) (method from Javanmard & Montanari (2014a)) would have taken more than 2 days, whereas our Fast Debiasing approach accomplishes it in less than a second.

**Computation Time:** In both cases above, we note that the fast debiasing approach allowed for construction of the debiasing matrix $\boldsymbol{W}$ in about a second. On the other hand, using the method from Javanmard & Montanari (2014a) to construct $\boldsymbol{M}$ would have required more than two days.

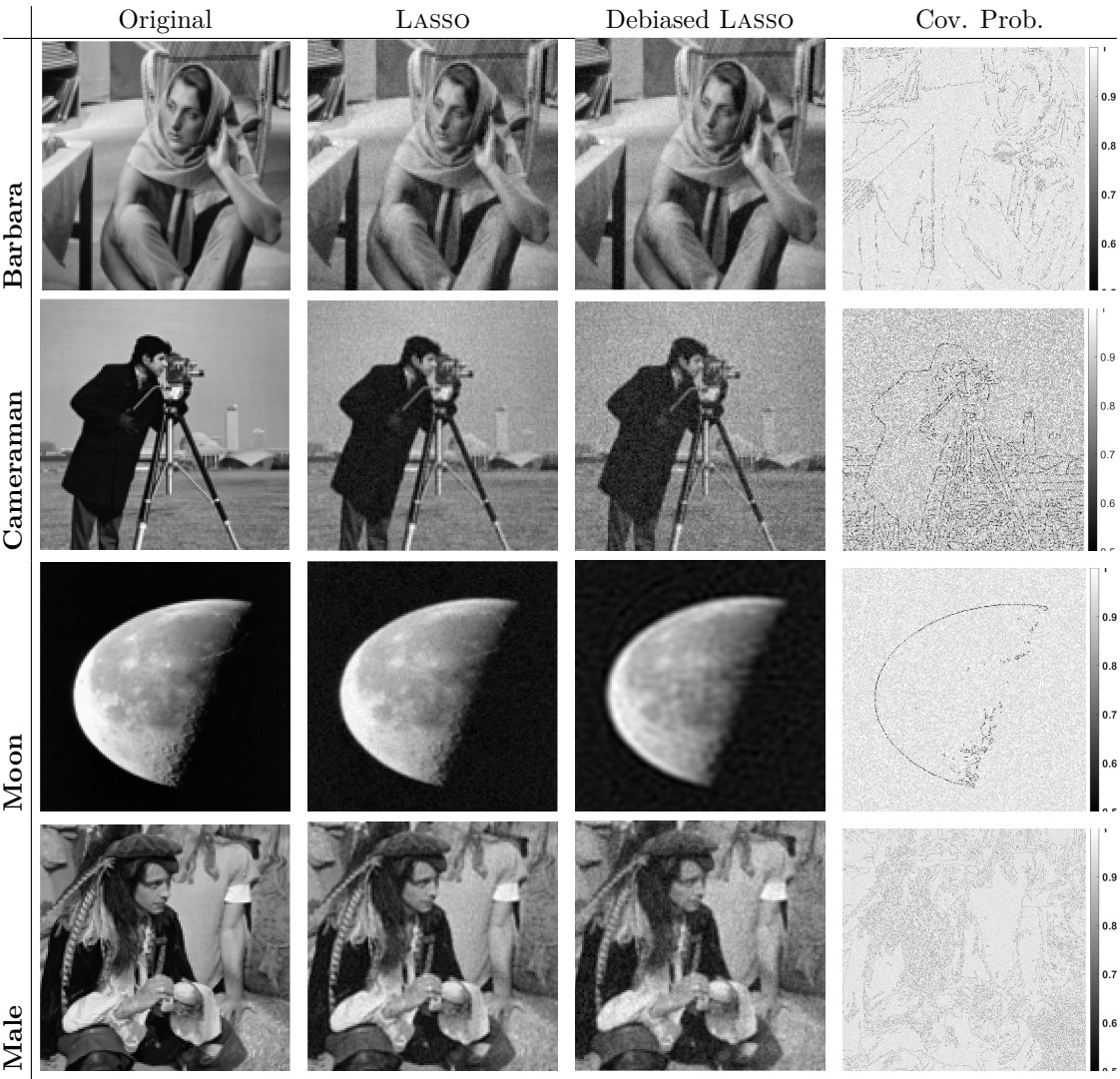

Figure 4: Compressive Image Reconstruction given cross-talk (correlation in the entries of the sensing matrix): Reconstruction of grayscale images of dimension $256 \times 256$ (leftmost column) using LASSO (column 2 from left) and Debiased LASSO (column 3 from left) and the empirical coverage probabilities $N_i/K \ \forall i \in [p]$ based on confidence intervals for the debiased LASSO estimates (rightmost column). Here $n = 20000, p = 65,536$, and the additive noise $\sigma$ is taken as 1% of mean absolute noiseless measurements and $\zeta = 0.02$. Debiasing matrix computation using (4) (method from Javanmard & Montanari (2014a)) would have taken more than 2 days, whereas our Fast Debiasing approach accomplishes it in less than a second.

## 6.2 Experiments on Compressive Video Reconstruction

Here, we follow the forward model of well known video compressed sensing architectures such as Liu et al. (2013), which acquire snapshot images representing the superposition of a set of pixel-wise modulated consecutive video frames. This snapshot image $\boldsymbol{y} \in \mathbb{R}^{p^2}$ (an image of size $p \times p$, reshaped to form a vector) is

|  | RRMSE | SSIM | Cov. Prob. (Edges) | Cov. Prob. (Non-Edges) |
|---|---|---|---|---|
| **Barbara** | 0.0445 | 0.901 | 0.65 | 0.90 |
| **Cameraman** | 0.0481 | 0.909 | 0.61 | 0.89 |
| **Moon** | 0.0317 | 0.923 | 0.72 | 0.92 |
| **Male** | 0.0457 | 0.898 | 0.58 | 0.89 |

Table 7: Reconstruction quality metrics for grayscale images of size $256 \times 256$ using Debiased LASSO given cross-talk (correlation in the entries of the sensing matrix) with $\zeta = 0.02$. The metrics reported include RRMSE, SSIM, and average coverage probabilities computed separately on edge and non-edge pixel locations based on the Canny edge detection technique.

represented in the form:

$$y = \sum_{t=1}^{T} \Phi_t \circ f_t + \eta, \tag{9}$$

where $\eta \in \mathbb{R}^{p^2}$ is a noise vector with elements drawn from $\mathcal{N}(0, \sigma^2)$, $\Phi_t \in \mathbb{R}^{p^2}$ is a randomly generated Bernoulli or Rademacher pattern for modulating frame $f_t$, the $t$-th frame of the underlying video. The aim is to reconstruct the video $f := \{f_t\}_{t=1}^{T}$ from $y$ and $\{\Phi_t\}_{t=1}^{T}$. The video is a 3D signal of size $p \times p \times T$, which after vectorization can be regarded as a vector in $\mathbb{R}^{Tp^2}$. The effective sensing matrix $\Phi$ (of size $p^2 \times Tp^2$) has the form $\Phi = (\text{diag}(\Phi_1)|\text{diag}(\Phi_2)|...|\text{diag}(\Phi_T))$, where $\text{diag}(\Phi_t)$ is a diagonal matrix of size $p^2 \times p^2$ containing the elements of $\Phi_t$ on its diagonal.

For our experiments, we represented the 3D signal $f \in \mathbb{R}^{p^2 T}$ in the 3D-DCT basis $\Psi_{3D}$ in the form $f = \Psi_{3D}\theta$ where $\theta \in \mathbb{R}^{p^2 T}$ is a (weakly) sparse vector of 3D DCT coefficients. The snapshot images were obtained by simulating the forward model on an already available video. The video was reconstructed by the LASSO using $\hat{\theta} := \arg\min_{\theta} \|y - \Phi\Psi_{3D}\theta\|_2^2 + \lambda\|\theta\|_1$, followed by a debiasing step where the debiasing matrix was computed using our fast approach. Reconstruction results for the first set of 5 frames are shown in Fig. 5—the supplemental material contains comparative reconstructions in video format. These reconstructions reveal good quality reconstruction of spatial textures as well as temporal motion patterns.

Here again, the aim of the experiment here is to show that the fast debiasing approach works for another compressive architecture where the sensing matrix $\Phi$ consists of a column-wise concatenation of diagonal sub-matrices. The method of Javanmard & Montanari (2014a) would have taken more than 2 days for computing the debiasing matrix, whereas our technique obtains $W$ in less than a second.

## 6.3 Experiments with Hyperspectral Image Reconstruction

We also experimented with reconstruction of hyperspectral images from *real* compressive acquisitions in the form of coded snapshot images acquired by the CASSI (Coded aperture snapshot spectral imager) camera Kittle et al. (2010). Consider a hyperspectral image $f$ of size $p_x \times p_y \times n_L$ where $n_L$ is the number of spectral channels (or wavelengths). We represent $f$ as a vector in $\mathbb{R}^{p_x p_y n_L}$. A hyperspectral image can be regarded as a stack of wavelength-specific slices, where each slice is an image of size $p_x \times p_y$. The CASSI camera does not measure the entire image $f$, but instead measures a coded snapshot image $y$ (roughly of size $p_x \times p_y$, represented as a vector in $\mathbb{R}^{p_x p_y}$) in the form of a superposition of the $L$ individual slices each modulated by binary patterns. This is expressed mathematically in the following manner:

$$y = \sum_{l=1}^{n_L} f_i \cdot C_i + \eta, \tag{10}$$

where $f_i$ (a vector in $\mathbb{R}^{p_x p_y}$) is the $i$th slice of $f$ and $C_i$ (a vector in $\{0,1\}^{p_x p_y}$) is the binary code for the $i$th slice. The binary codes are implemented in hardware via a coded aperture (or mask) that modulates the white light entering the camera. A prism inside the camera disperses this light into its constituent wavelengths and also gives rises to different shifts to each wavelength, ensuring that each slice is modulated

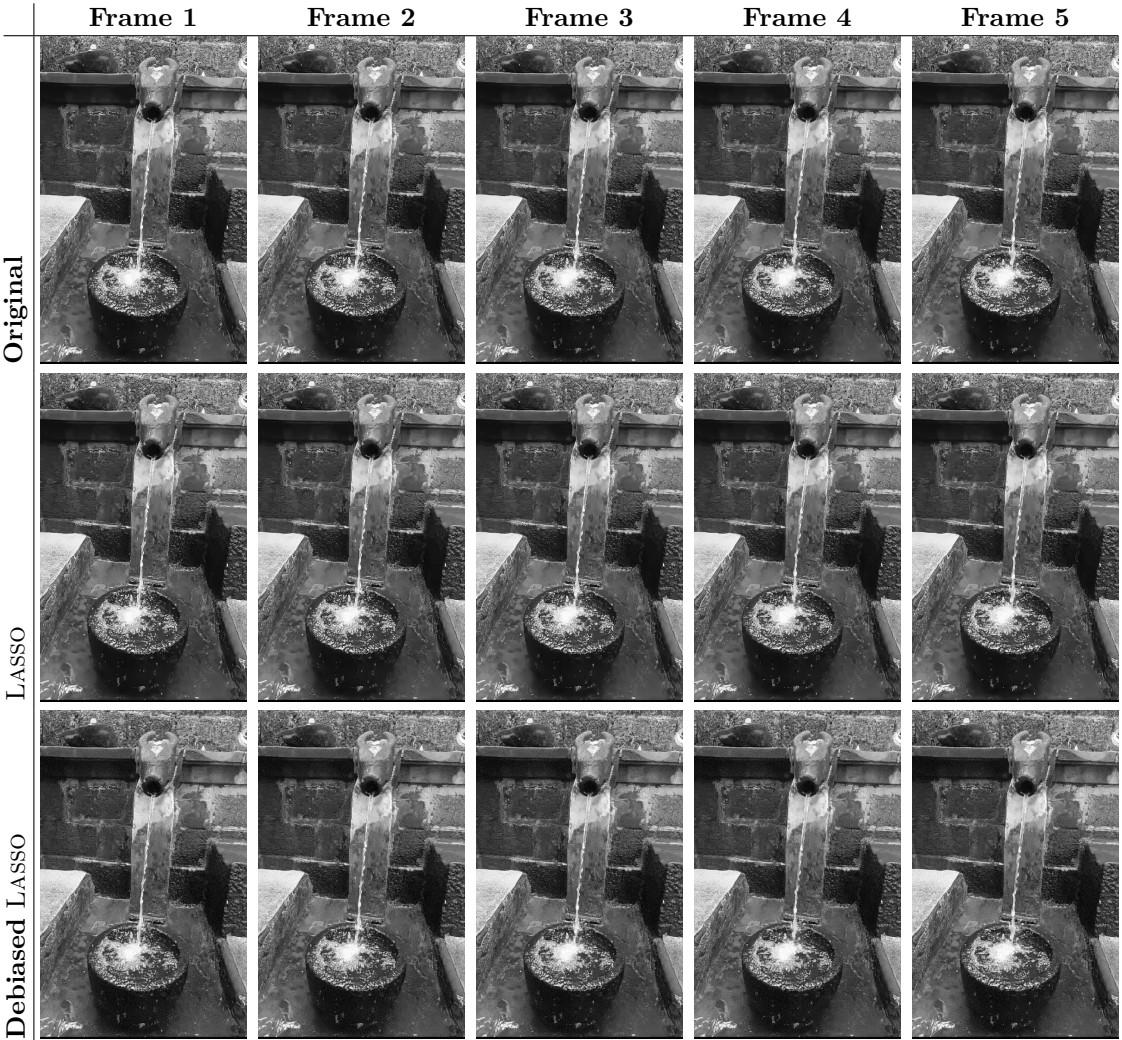

Figure 5: Video reconstruction for a water fountain scene: Reconstruction results showing the progression from original images via LASSO and Debiased LASSO. Each column corresponds to an individual frame and the first row corresponds to 5 frames of the original video, the second row shows the frames reconstructed using LASSO, the third row shows the frames reconstructed using Debiased LASSO. The compressed measurements correspond to $T = 5$ in (9). The average RRMSE and SSIM over the 5 frames are given as 0.0189 and 0.9293 respectively. Debiasing matrix computation using (4) (method from Javanmard & Montanari (2014a)) would have taken more than 2 days, whereas Fast Debiasing accomplishes it in less than a second. See supplemental video for results in video format.

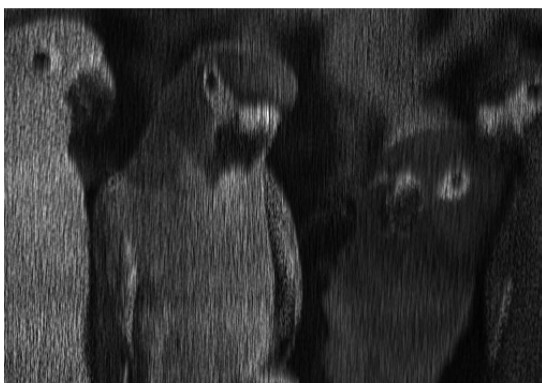

Figure 6: Coded snapshot used for hyperspectral reconstruction in Fig. 7.

by a different binary code before the superposition of the individual slices is recorded by the sensor. For more details, see Kittle et al. (2010). The aim is to reconstruct $\boldsymbol{f}$ from $\boldsymbol{y}$ and $\{\boldsymbol{C}_i\}_{i=1}^{n_L}$. Since the associated compression ratio here ($n_L : 1$) is very high, in practice a multi-snapshot version of CASSI is used, where $T < n_L$ *different* coded snapshot images are acquired, each with a *different* coded aperture pattern. The forward model now is:

$$\forall t \in \{1, 2, ..., T\}, \boldsymbol{y}_t = \sum_{l=1}^{n_L} \boldsymbol{f}_i \cdot \boldsymbol{C}_{it} + \boldsymbol{\eta}_t, \tag{11}$$

where $\boldsymbol{y}_t$ is the $t$-th coded snapshot, $\boldsymbol{C}_{it}$ (a vector in $\{0,1\}^{p_x p_y}$) is the coded aperture pattern for the $i$th channel in the $t$-th snapshot. The aim is to reconstruct $\boldsymbol{f}$ from $\{\boldsymbol{y}_t\}_{t=1}^T$ and $\{\{\boldsymbol{C}_{i,t}\}_{t=1}^T\}_{i=1}^{n_L}$. Here again, we employ a Lasso estimator with a 3D-DCT representation for $\boldsymbol{f} = \boldsymbol{\Psi}\boldsymbol{\theta}$ where $\boldsymbol{\Psi} \in \mathbb{R}^{p_x p_y n_L \times p_x p_y n_L}$ is the 3D-DCT basis matrix and $\boldsymbol{\theta} \in \mathbb{R}^{p_x p_y n_L}$ is a vector of 3D-DCT coefficients. The estimate of $\boldsymbol{\theta}$ is obtained by minimizing $\|\boldsymbol{z} - \boldsymbol{\Phi}\boldsymbol{\Psi}\boldsymbol{\theta}\|_2^2 + \lambda\|\boldsymbol{\theta}\|_1$ where $\boldsymbol{z}$ is a vector of size $T p_x p_y \times 1$ obtained by concatenating all the vectorized coded snapshots $\{\boldsymbol{y}_t\}_{t=1}^T$ and $\boldsymbol{\Phi}$ is a matrix of size $T p_x p_y \times n_L p_x p_y$ defined as follows:

$$\boldsymbol{z} := \begin{pmatrix} \boldsymbol{y}_1 \\ . \\ . \\ . \\ \boldsymbol{y}_T \end{pmatrix}, \boldsymbol{\Phi} := \begin{pmatrix} \mathrm{diag}(\boldsymbol{C}_{1,1}) & \mathrm{diag}(\boldsymbol{C}_{2,1}) & \ldots & \mathrm{diag}(\boldsymbol{C}_{n_L,1}) \\ . & . & . & . \\ . & . & . & . \\ . & . & . & . \\ \mathrm{diag}(\boldsymbol{C}_{1,T}) & \mathrm{diag}(\boldsymbol{C}_{2,T}) & \ldots & \mathrm{diag}(\boldsymbol{C}_{n_L,T}) \end{pmatrix}, \tag{12}$$

where $\forall i \in [n_L], \forall t \in [T]$, $\mathrm{diag}(\boldsymbol{C}_{i,t})$ denotes a diagonal matrix of size $p_x p_y \times p_x p_y$ containing the $p_x p_y$ elements of $\boldsymbol{C}_{i,t}$ along its diagonal.

For our experiments, we used a set of $T = 6$ coded snapshot images acquired by a *real* camera, corresponding to a hyperspectral image of size $1021 \times 730 \times 24$ with $n_L = 24$. One of these snapshot images is shown in Fig. 6. Four different slices of the reconstructed hypercube using Lasso and debiased Lasso (using our fast approach for computing the debiasing matrix) are shown in Fig. 7. Since ground truth is absent, RRMSE or SSIM values cannot be directly computed. However the reconstruction with $T = 6$ (compression ratio of 4:1) snapshots can be compared to a reconstruction with $T = 24$ snapshots (compression ratio 1:1). The latter can be regarded as a form of ground truth since there is no compression. We observe that the debiased reconstructions with $T = 6$ snapshots quite closely match those with $T = 24$, validating the success of our debiasing approach on this architecture as well. The RRMSE and SSIM values between reconstructions under $T = 6$ and $T = 24$ averaged over all bands are 0.0023 and 0.9912 for Lasso and 0.0029 and 0.9891 for Debiased Lasso respectively.

## 7   Conclusion

In this article, we reformulate the optimization problem to obtain $\boldsymbol{M}$ (the approximate inverse of the covariance matrix of the rows of the sensing matrix $\boldsymbol{A}$) in Javanmard & Montanari (2014a) and further provide

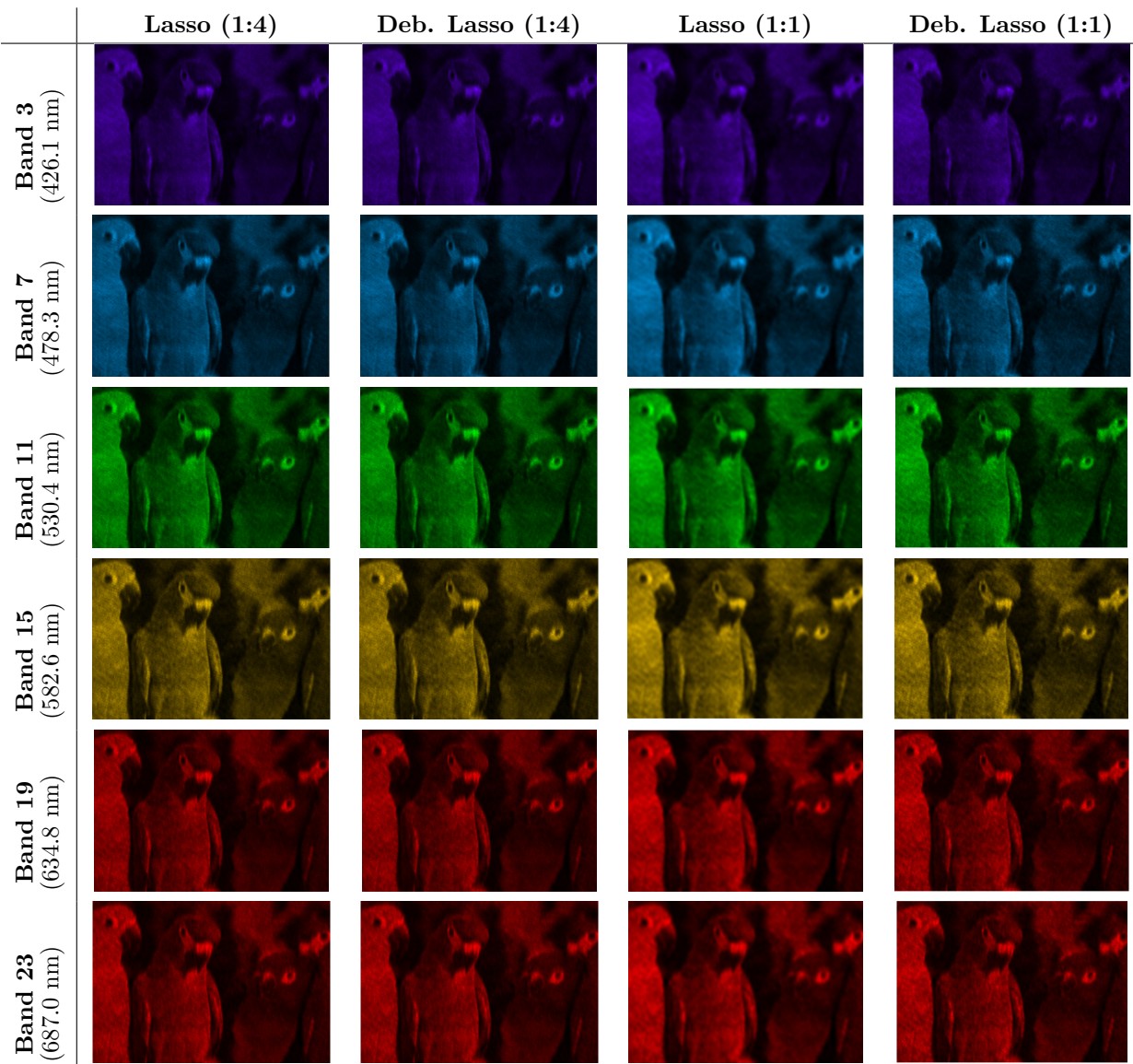

Figure 7: Reconstruction of a hyperspectral image of size $1021 \times 730 \times 24$: Rows correspond to bands 3, 7, 11, 15, 19, and 23 (wavelengths 426.1–687.0 nm). Columns correspond to reconstructed images using LASSO and Debiased LASSO reconstructions under 1:4 (that is, with $T = 6$ different coded snapshots in (11)) and 1:1 (that is, with $T = 24$ different coded snapshots in (11)) compression ratios respectively. The slices at any wavelength are represented using the color corresponding to the wavelength. The RRMSE and SSIM values between reconstructions under 1:4 and 1:1 averaged over all bands are 0.0023 and 0.9912 for LASSO and 0.0029 and 0.9891 for Debiased LASSO respectively.

an exact, closed-form optimal solution to the reformulated problem under assumptions on the pairwise inner products of the columns of $A$. For sensing matrices with i.i.d. zero-mean sub-Gaussian rows that have a diagonal covariance matrix or a full covariance matrix with small-valued off-diagonal elements, the debiased LASSO estimator, based on this closed-form solution, has entries that are asymptotically zero-mean and sub-Gaussian. The exact solution significantly improves the time efficiency for debiasing the LASSO estimator, as shown in the numerical results. Our method is particularly useful for debiasing in streaming settings where new measurements or new signal features arrive on the fly.

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

# A    Appendix for 'Fast Debiasing of the Lasso Estimator': Proofs of Theoretical Results

## A.1    Proof of Theorem 1

**Primal feasibility:**   If $\frac{\rho}{1+\rho} \leq \mu \leq 1$ then we have that $\mu + \mu\rho \geq \rho$ which implies that $0 \leq (1-\mu)\rho \leq \mu$. The choice of $\boldsymbol{w}_{\cdot j}$ given by (7) is primal feasible since

$$\left\| \frac{1}{n} \boldsymbol{A}^\top \frac{(1-\mu)}{\frac{\|\boldsymbol{a}_{\cdot j}\|_2^2}{n}} \boldsymbol{a}_{\cdot j} - \boldsymbol{e}_j \right\|_\infty \leq \max\{\mu, |(1-\mu)\rho|\} = \mu. \tag{13}$$

To see why this is true, note that for index $j$, the LHS is upper bounded by $\mu$, otherwise it is upper bounded by $|(1-\mu)\rho|$.

**Primal objective function value:**   The primal objective function value is given by $\frac{1}{n}\|\boldsymbol{w}_{\cdot j}\|_2^2 = \frac{(1-\mu)^2}{(\|\boldsymbol{a}_{\cdot j}\|_2^2/n)^2}\|\boldsymbol{a}_{\cdot j}\|_2^2/n = \frac{(1-\mu)^2}{\|\boldsymbol{a}_{\cdot j}\|_2^2/n}$.

**The Fenchel dual problem:**   Consider an optimization problem of the form for a fixed $j \in [p]$:

$$\inf_{\boldsymbol{w}} f(\boldsymbol{w}) + g_j\left(\frac{1}{n}\boldsymbol{A}^\top \boldsymbol{w}\right) \tag{14}$$

where $f$ and $g_j$ are extended real-valued convex functions. The Fenchel dual (see Chapter 3 of Borwein & Lewis (2006)) is

$$\sup_{\boldsymbol{u}} -f^*\left(\frac{1}{n}\boldsymbol{A}\boldsymbol{u}\right) - g_j^*(-\boldsymbol{u}) \tag{15}$$

where $f^*$ and $g_j^*$ are the convex conjugates of $f$ and $g_j$ respectively. The Fenchel dual satisfies weak duality (see Chapter 3 of Borwein & Lewis (2006)), i.e., for any $\boldsymbol{w}$ and $\boldsymbol{u}$,

$$f(\boldsymbol{w}) + g_j\left(\frac{1}{n}\boldsymbol{A}^\top \boldsymbol{w}\right) \geq -f^*\left(\frac{1}{n}\boldsymbol{A}\boldsymbol{u}\right) - g_j^*(-\boldsymbol{u}).$$

In our setting, for a fixed $j$, we consider

$$f(\boldsymbol{w}) := \frac{1}{n}\|\boldsymbol{w}\|^2 \quad \text{and} \quad g_j(\boldsymbol{w}) := \begin{cases} 0 & \text{if } \|\boldsymbol{w} - \boldsymbol{e}_j\|_\infty \leq \mu \\ \infty & \text{otherwise} \end{cases}. \tag{16}$$

Then, for the same $j$, we have their convex conjugates from Lemma 3:

$$f^*(\boldsymbol{u}) = \sup_{\boldsymbol{w}} \boldsymbol{u}^\top \boldsymbol{w} - f(\boldsymbol{w}) = \frac{n}{4}\|\boldsymbol{u}\|^2, \tag{17}$$

$$g_j^*(\boldsymbol{u}) = \sup_{\boldsymbol{w}} \boldsymbol{u}^\top \boldsymbol{w} - g_j(\boldsymbol{w}) = \sup_{\|\boldsymbol{w}-\boldsymbol{e}_j\|_\infty \leq \mu} \boldsymbol{u}^\top \boldsymbol{w} = u_j + \mu\|\boldsymbol{u}\|_1. \tag{18}$$

This gives a dual problem in the form $\sup_{\boldsymbol{u}} -\frac{1}{4n}\boldsymbol{u}^\top \boldsymbol{A}^\top \boldsymbol{A}\boldsymbol{u} + u_j - \mu\|\boldsymbol{u}\|_1$.

The point $\boldsymbol{u} := \dfrac{2(1-\mu)\boldsymbol{e}_j}{\|\boldsymbol{a}_{.j}\|_2^2/n}$ is feasible for the dual (trivially, as there are no constraints).

**Dual objective function value:** Plugging in $\boldsymbol{u} = \dfrac{2(1-\mu)\boldsymbol{e}_j}{\|\boldsymbol{a}_{.j}\|_2^2/n}$, the corresponding dual objective function value is

$$
\begin{aligned}
-\frac{1}{4n}\boldsymbol{u}^\top \boldsymbol{A}^\top \boldsymbol{A}\boldsymbol{u} + u_j - \mu\|\boldsymbol{u}\|_1 &= -\frac{1}{4n}\|\boldsymbol{a}_{.j}\|^2 \frac{4(1-\mu)^2}{(\|\boldsymbol{a}_{.j}\|_2^2/n)^2} + \frac{2(1-\mu)}{\|\boldsymbol{a}_{.j}\|_2^2/n} - \mu\frac{2(1-\mu)}{\|\boldsymbol{a}_{.j}\|_2^2/n} \\
&= -\frac{(1-\mu)^2}{\|\boldsymbol{a}_{.j}\|_2^2/n} + 2\frac{(1-\mu)^2}{\|\boldsymbol{a}_{.j}\|_2^2/n} = \frac{(1-\mu)^2}{\|\boldsymbol{a}_{.j}\|_2^2/n}.
\end{aligned}
$$

Since the primal solution and the dual objective function values are equal, it follows that an optimal solution for the primal is $\dfrac{(1-\mu)}{\|\boldsymbol{a}_{.j}\|_2^2/n}\boldsymbol{a}_{.j}$, and that an optimal solution to the dual is $\dfrac{2(1-\mu)}{\|\boldsymbol{a}_{.j}\|_2^2/n}\boldsymbol{e}_j$.

We have shown that if $\rho/(1+\rho) \leq \mu \leq 1$ then the optimal solution of (6) is given by (7). Now consider the case when $\mu < \rho/(1+\rho)$. This implies $\mu < (1-\mu)\rho$. Let $i,j \in [p]$ (with $i \neq j$) be such that $\rho = |\boldsymbol{a}_{.i}^\top \boldsymbol{a}_{.j}|/\|\boldsymbol{a}_{.j}\|_2^2$. Then plugging in the expression $\boldsymbol{w}_{.j} := \frac{n(1-\mu)}{\|\boldsymbol{a}_{.j}\|_2^2}\boldsymbol{a}_{.j}$ from (7) into the constraint of (6) we have,

$$
\left\| \frac{1}{n}\boldsymbol{A}^\top \frac{(1-\mu)}{\frac{\|\boldsymbol{a}_{.j}\|_2^2}{n}}\boldsymbol{a}_{.j} - \boldsymbol{e}_j \right\|_\infty \geq (1-\mu)|\boldsymbol{a}_{.i}^\top \boldsymbol{a}_{.j}|/\|\boldsymbol{a}_{.j}\|_2^2 = (1-\mu)\rho > \mu.
$$

This shows that $\boldsymbol{w}_{.j}$ (defined in (7)) is not feasible for (6) when $\mu < \rho/(1+\rho)$, and so is certainly not optimal.

Finally, consider the case when $\mu > 1$. If $\mu \geq 1$, then the unique optimal solution of (6) is $\boldsymbol{w}_{.j} = \boldsymbol{0}$. This is because $\boldsymbol{0}$ is feasible and is the global minimizer of the objective function. However, when $\mu > 1$, the formula (7) does not give the value $\boldsymbol{0}$, and so is not the optimal solution to (6).

This concludes the proof that $\frac{\rho}{1+\rho} \leq \mu \leq 1$ is necessary and sufficient condition for the expression given in (7) to be optimal.

## A.2 Proof of Theorem 2

For an $n \times p$ matrix $\boldsymbol{A}$, let for all $j \in [p]$,

$$
L_j := \frac{1}{n}\|\boldsymbol{a}_{.,j}\|_2^2 \tag{19}
$$

and let for all $l \neq j \in [p]$,

$$
\nu_{lj} := \frac{1}{n}|\boldsymbol{a}_{.l}^\top \boldsymbol{a}_{.j}|. \tag{20}
$$

Using union bound on (32) of Lemma 2 and (31) of Lemma 1, we have under the assumption $n \geq \frac{8C_{\max}^2\kappa^4}{C_{\min}^2(1-c)^2}\log p$ for all $l \neq j \in [p]$,

$$
P\left( \frac{\nu_{lj}}{L_j} \geq \frac{2\sqrt{2}C_{\max}\kappa^2\sqrt{\frac{\log p}{n}} + |\Sigma_{lj}|}{c\Sigma_{jj}} \right) \leq \frac{3}{p^4}. \tag{21}
$$

Given the definition of $\rho$ in Theorem 1, we have the bound

$$
\frac{\rho}{1+\rho} \leq \rho = \max_{l \neq j}\frac{|\boldsymbol{a}_{.l}^\top \boldsymbol{a}_{.j}|}{\|\boldsymbol{a}_{.j}\|_2^2} = \max_{l \neq j}\frac{\nu_{lj}}{L_j}. \tag{22}
$$

Taking union bound over $l \neq j \in [p]$, we have,

$$P\left(\frac{\rho}{1+\rho} \leq \max_{l \neq j} \frac{\nu_{lj}}{L_j} \leq \max_{l \neq j} \frac{2\sqrt{2}C_{\max}\kappa^2\sqrt{\frac{\log p}{n}}}{c\Sigma_{jj}} + \frac{1}{c}\max_{l \neq j}\frac{|\Sigma_{lj}|}{\Sigma_{jj}}\right) \geq 1 - \frac{p(p-1)}{2}\frac{3}{p^4} \geq 1 - \frac{3}{2p^2}. \tag{23}$$

Since $\Sigma_{jj} \geq C_{\min}$ for all $j \in [p]$, we have, $\max_{l \neq j} \dfrac{2\sqrt{2}C_{\max}\kappa^2\sqrt{\frac{\log p}{n}}}{c\Sigma_{jj}} \leq \dfrac{2\sqrt{2}C_{\max}\kappa^2\sqrt{\frac{\log p}{n}}}{cC_{\min}}$. Furthermore,

given $\gamma \geq \dfrac{C_{\min}}{\kappa^2 C_{\max}}\sqrt{\frac{n}{\log p}}\max_{l \neq j}\dfrac{|\Sigma_{lj}|}{\Sigma_{jj}}$, we have,

$$P\left(\frac{\rho}{1+\rho} \leq (2\sqrt{2}+\gamma)\frac{\kappa^2}{c}\frac{C_{\max}}{C_{\min}}\sqrt{\frac{\log p}{n}}\right) \geq 1 - \frac{3}{2p^2}.$$

We have now established the upper bound on $\rho/(1+\rho)$ with high probability. Theorem 1 states that for $\rho/(1+\rho) \leq \mu < 1$ the optimization problem in (6) is feasible and the optimal debiasing matrix $\boldsymbol{W}$ in (6) is given by (7). The choice $\mu = (2\sqrt{2}+\gamma)\frac{\kappa^2}{c}\frac{C_{\max}}{C_{\min}}\sqrt{\frac{\log p}{n}}$ with $c \in \left(\frac{2\sqrt{2}+\gamma}{4\sqrt{2}+\gamma}, 1\right)$ ensures that $\mu < 1$ and $\rho/(1+\rho) \leq \mu$ with high probability.

This completes the proof of Theorem 2.

### A.3  Lower bound on $L_j$

In Lemma 1, we show that for an ensemble of sensing matrices satisfying assumptions **D1**, **D2**, the parameter $L_j$ is greater than $c\Sigma_{jj}$ for all $j \in [p]$, with high probability, for some constant $c$.

**Lemma 1** *Let $\boldsymbol{A}$ be a $n \times p$ matrix with independently and identically distributed sub-Gaussian rows, where $n < p$. Consider $L$ as defined in (19). For any constant $c \in (0,1)$ and $\kappa := \|\boldsymbol{\Sigma}^{-1/2}\boldsymbol{a_{i.}}\|_{\psi_2}$, if $\boldsymbol{A}$ satisfies properties $\boldsymbol{D1}$ and $\boldsymbol{D2}$ and $n \geq \frac{8C_{\max}^2\kappa^4}{C_{\min}^2(1-c)^2}\log p$, then for all $j \in [p]$,*

$$P\left(L_j \geq c\,\Sigma_{jj}\right) \geq 1 - \frac{2}{p^4}. \tag{24}$$

**Proof of Lemma 1** We have for all $j \in [p]$, $\frac{\|\boldsymbol{a_{.j}}\|_2^2}{n} = \frac{1}{n}\sum_{i=1}^{n}a_{ij}^2$. Since the $\boldsymbol{a_{i.}}$ (for $i \in [n]$) are sub-Gaussian, the $a_{ij}$ are sub-Gaussian for each $j \in [p]$ and $\|a_{ij}\|_{\psi_2} \leq \|\boldsymbol{a_{i.}}\|_{\psi_2}$. By the definition of the sub-Gaussian norm (see footnote in Sec. 2 with $q = 2$), we know that

$$\frac{1}{2}E[a_{ij}^2] \leq \|a_{ij}\|_{\psi_2}^2 = \|\boldsymbol{e}_j^\top \boldsymbol{a_{i.}}\|_{\psi_2}^2 \leq \|\boldsymbol{a_{i.}}\|_{\psi_2}^2. \tag{25}$$

Recall that $\kappa := \|\boldsymbol{\Sigma}^{-1/2}\boldsymbol{a_{i.}}\|_{\psi_2}$ in property **D1** of sensing matrix $\boldsymbol{A}$. We have

$$\begin{aligned}
\|\boldsymbol{a_{i.}}\|_{\psi_2} &= \sup_{\boldsymbol{v} \in S^{p-1}} \left\|(\boldsymbol{\Sigma}^{1/2}\boldsymbol{v})^\top\boldsymbol{\Sigma}^{-1/2}\boldsymbol{a_{i.}}\right\|_{\psi_2} \\
&= \sup_{\boldsymbol{v} \in S^{p-1}} \|\boldsymbol{\Sigma}^{1/2}\boldsymbol{v}\|_2 \left\|\frac{1}{\|\boldsymbol{\Sigma}^{1/2}\boldsymbol{v}\|_2}(\boldsymbol{\Sigma}^{1/2}\boldsymbol{v})^\top\boldsymbol{\Sigma}^{-1/2}\boldsymbol{a_{i.}}\right\|_{\psi_2} \\
&\leq \sup_{\boldsymbol{v} \in S^{p-1}} \|\boldsymbol{\Sigma}^{1/2}\boldsymbol{v}\|_2 \sup_{\boldsymbol{z} \in S^{p-1}} \left\|\frac{1}{\|\boldsymbol{\Sigma}^{1/2}\boldsymbol{z}\|_2}(\boldsymbol{\Sigma}^{1/2}\boldsymbol{z})^\top\boldsymbol{\Sigma}^{-1/2}\boldsymbol{a_{i.}}\right\|_{\psi_2} \\
&\leq \sigma_{\max}(\boldsymbol{\Sigma}^{1/2})\|\boldsymbol{\Sigma}^{-1/2}\boldsymbol{a_{i.}}\|_{\psi_2} \\
&\leq \sqrt{C_{\max}}\,\kappa,
\end{aligned} \tag{26}$$

where $C_{\max}$ is defined in property **D2**. Therefore, we obtain $E[a_{ij}^2] \leq 2\|\boldsymbol{a}_{i\cdot}\|_{\psi_2}^2 \leq 2C_{\max}\kappa^2$. From the definition of eigenvalues, for any $\boldsymbol{x} \in \mathbb{R}^p$, $\boldsymbol{x}^\top \boldsymbol{\Sigma} \boldsymbol{x} \geq \sigma_{\min}(\boldsymbol{\Sigma})\|\boldsymbol{x}\|_2^2 \geq C_{\min}\|\boldsymbol{x}\|_2^2$. Putting $\boldsymbol{x} = \boldsymbol{e}_j$, where $\boldsymbol{e}_j$ is the $j^{\text{th}}$ column of $\boldsymbol{I_p}$, we have, $\Sigma_{jj} \geq C_{\min}$. Since $E[a_{ij}^2] = \Sigma_{jj} \geq C_{\min}$, we have, $E\left[\frac{1}{n}\sum_{i=1}^n a_{ij}^2\right] \geq C_{\min}$.

For a given $j \in [p]$, the variables $a_{ij}^2$ are independent for all $i \in [n]$. Hence, using the concentration inequality of Theorem 3.1.1 and Equation (3.3) of Vershynin (2018), we have for $t > 0$[2],

$$P\left(\left|\|\boldsymbol{a}_{\cdot j}\|_2^2/n - E[\|\boldsymbol{a}_{\cdot j}\|_2^2/n]\right| \geq t\right) \leq 2e^{-\frac{nt^2}{2C_{\max}^2\kappa^4}}. \tag{27}$$

Using the left-sided inequality of (27), we have,

$$P\left(\|\boldsymbol{a}_{\cdot j}\|_2^2/n \leq E[\|\boldsymbol{a}_{\cdot j}\|_2^2/n] - t\right) \leq 2e^{-\frac{nt^2}{2C_{\max}^2\kappa^4}}. \tag{28}$$

Using $E[\|\boldsymbol{a}_{\cdot j}\|_2^2/n] = \Sigma_{jj}$, (28) can be rewritten as follows for $t > 0$:

$$P\left(L_j \leq \Sigma_{jj} - t\right) \leq 2e^{-\frac{nt^2}{2C_{\max}^2\kappa^4}}. \tag{29}$$

Putting $t := 2\sqrt{2}C_{\max}\kappa^2\sqrt{\frac{\log p}{n}}$ in (29), we obtain:

$$P\left(L_j \leq \Sigma_{jj}\left(1 - 2\sqrt{2}\frac{C_{\max}}{\Sigma_{jj}}\kappa^2\sqrt{\frac{\log p}{n}}\right)\right) \leq \frac{2}{p^4}. \tag{30}$$

For some constant $c \in (0, 1)$, if $n \geq \frac{8C_{\max}^2\kappa^4}{C_{\min}^2(1-c)^2}\log p \geq \frac{8C_{\max}^2\kappa^4}{\Sigma_{jj}^2(1-c)^2}\log p$ for all $j \in [p]$, then (30) becomes:

$$P(L_j \leq c\Sigma_{jj}) \leq \frac{2}{p^4} \implies P(L_j \geq c\Sigma_{jj}) \geq 1 - \frac{2}{p^4}. \tag{31}$$

This completes the proof.

### A.4  Upper bound on $\nu_{lj}$

In the upcoming Lemma we provide a high probability upper bound on $\nu_{lj} \, \forall \, l \neq j \in [p]$, for sensing matrices with independent and identically distributed zero-mean sub-Gaussian rows.

**Lemma 2** *Let $\boldsymbol{A}$ be a $n \times p$ dimensional matrix satisfying assumptions **D1** and **D2** and with sub-Gaussian norm $\kappa := \|\boldsymbol{\Sigma}^{-1/2}\boldsymbol{a_{i\cdot}}\|_{\psi_2}$. Define $\nu_{lj}$ as in (20). Then for all $l \neq j \in [p]$,*

$$P\left(\nu_{lj} \leq 2\sqrt{2}C_{\max}\kappa^2\sqrt{\frac{\log p}{n}} + |\Sigma_{lj}|\right) \geq 1 - \frac{1}{p^4}. \tag{32}$$

**Proof of Lemma 2** We have $\frac{1}{n}|\boldsymbol{a}_{\cdot l}^\top \boldsymbol{a}_{\cdot j}| = \frac{1}{n}\sum_{i=1}^n a_{ij}a_{il}$. Here, for given $j \neq l$, we know that $a_{ij}$ and $a_{il}$ are independent zero-mean sub-Gaussian random variables. From from (25) and (26) we know that their sub-Gaussian norm is at most $\sqrt{C_{\max}}\kappa$ for all $i \in [n]$. Using Lemma 2.7.7 of Vershynin (2018), we have that for all $i \in [n]$, $a_{ij}a_{il}$ are independent sub-Exponential random variables with sub-exponential norm at most $C_{\max}\kappa^2$. Moreover, $E[a_{ij}a_{il}] = \boldsymbol{\Sigma}_{jl}$. Hence, using Bernstein's inequality for averages of independent sub-exponential random variables, given in Corollary 2.8.3 of Vershynin (2018), we have for any $t > 0$,

$$P\left(\left|\frac{1}{n}\sum_{i=1}^n a_{ij}a_{il} - \frac{1}{n}\sum_{i=1}^n E[a_{ij}a_{il}]\right| \geq t\right) = P\left(\left|\frac{1}{n}\sum_{i=1}^n a_{ij}a_{il} - \Sigma_{lj}\right| \geq t\right) \leq 2e^{-\frac{nt^2}{2C_{\max}^2\kappa^4}} \tag{33}$$

---

[2]We have set $c = 1/2$, $\delta := t$ and $K := 2\sqrt{C_{\max}}\kappa$ in Equation (3.3) and the equation immediately preceding it in Vershynin (2018)

Using Reverse Triangle's inequality, we have,

$$\left| \frac{1}{n} \sum_{i=1}^{n} a_{ij} a_{il} \right| - |\Sigma_{jj}| \leq \left| \frac{1}{n} \sum_{i=1}^{n} a_{ij} a_{il} - \Sigma_{jj} \right|.$$

Thereofore, $\{ \left| \frac{1}{n} \sum_{i=1}^{n} a_{ij} a_{il} \right| - |\Sigma_{jj}| \geq t \} \implies \{ \left| \frac{1}{n} \sum_{i=1}^{n} a_{ij} a_{il} - \Sigma_{lj} \right| \}$. Hence, we have,

$$P\left( \left| \frac{1}{n} \sum_{i=1}^{n} a_{ij} a_{il} \right| \geq t + |\Sigma_{lj}| \right) \leq 2e^{-\frac{nt^2}{2C_{\max}^2 \kappa^4}} \tag{34}$$

Taking $t = 2\sqrt{2} C_{\max} \kappa^2 \sqrt{\frac{\log p}{n}}$, we have for all $l \neq j \in [p]$,

$$P\left( \nu_{lj} \geq 2\sqrt{2} C_{\max} \kappa^2 \sqrt{\frac{\log p}{n}} + |\Sigma_{lj}| \right) \leq \frac{1}{p^4}. \tag{35}$$

This completes the proof.

### A.5   Convex conjugates

The convex conjugate of a function $f(\boldsymbol{w})$ is defined as:

$$f^*(\boldsymbol{u}) = \sup_{\boldsymbol{w}} \left( \boldsymbol{u}^\top \boldsymbol{w} - f(\boldsymbol{w}) \right). \tag{36}$$

The following result gives the convex conjugates of the functions needed in the proof of Theorem 1.

**Lemma 3**      *1. If $f(\boldsymbol{w}) = \frac{1}{n}\|\boldsymbol{w}\|_2^2$, then its convex conjugate is $f^*(\boldsymbol{u}) = \frac{n}{4}\|\boldsymbol{u}\|_2^2$.*

   *2. If $g_j$ is the indicator function of the convex set $\{\boldsymbol{w} \in \mathbb{R}^p \mid \|\boldsymbol{w} - \boldsymbol{e_j}\|_\infty \leq \mu\}$, i.e.,*

$$g_j(\boldsymbol{w}) = \begin{cases} 0 & \text{if } \|\boldsymbol{w} - \boldsymbol{e_j}\|_\infty \leq \mu \\ \infty & \text{otherwise,} \end{cases}$$

   *then its convex conjugate is $g_j^*(\boldsymbol{u}) = u_j + \mu\|\boldsymbol{u}\|_1$.*

**Proof of Lemma 3:**

1. We can write $f(\boldsymbol{w}) = \frac{1}{2}\boldsymbol{w}^\top \boldsymbol{Q}\boldsymbol{w}$ where $\boldsymbol{Q} := \frac{2}{n}\boldsymbol{I_p}$ is positive definite (and has size $p \times p$). From Example 3.2.2 of Boyd & Vandenberghe (2004), the convex conjugate of a positive definite quadratic form is

$$f^*(\boldsymbol{u}) = \frac{1}{2}\boldsymbol{u}^\top \boldsymbol{Q}^{-1}\boldsymbol{u} = \frac{1}{2}\boldsymbol{u}^\top \left( \frac{2}{n}\boldsymbol{I_p} \right)^{-1} \boldsymbol{u} = \frac{n}{4}\|\boldsymbol{u}\|_2^2.$$

2. If $g_j$ is the indicator function of the set $C$, the convex conjugate is given by

$$g_j^*(\boldsymbol{u}) = \sup_{\boldsymbol{w} \in C} \boldsymbol{u}^\top \boldsymbol{w}, \tag{37}$$

   where $C = \{\boldsymbol{w} \in \mathbb{R}^p \mid \|\boldsymbol{w} - \boldsymbol{e_j}\|_\infty \leq \mu\}$. This implies that $w_i \in [e_{ji} - \mu, e_{ji} + \mu], \forall i$. (Note that $e_{ij} = 1$ if $i = j$ and 0 otherwise.) To maximize $\boldsymbol{u}^\top \boldsymbol{w} = \sum_{i=1}^{p} u_i w_i$, the optimal $w_i$ can be chosen as

$$w_i = \begin{cases} e_{ji} + \mu & \text{if } u_i \geq 0, \\ e_{ji} - \mu & \text{if } u_i < 0. \end{cases} \tag{38}$$

   Substituting into $\boldsymbol{u}^\top \boldsymbol{w}$, we obtain $\boldsymbol{u}^\top \boldsymbol{w} = \sum_{i=1}^{p} u_i \left( e_{ji} + \mu \operatorname{sign}(u_i) \right)$, where $\operatorname{sign}(u_i)$ is the sign of $u_i$. Simplifying, we have $\boldsymbol{u}^\top \boldsymbol{w} = u_j + \mu \sum_{i=1}^{p} |u_i|$. Thus, we have

$$g_j^*(\boldsymbol{u}) = u_j + \mu\|\boldsymbol{u}\|_1. \tag{39}$$

This completes the proof.

