# OpenReview forum: "Fast Debiasing of the LASSO Estimator"
_TMLR — Accepted by TMLR_

### Review · Reviewer_1iZg · 2025-10-12

**Summary Of Contributions:**

This paper presents a computationally efficient reformulation of the classic debiased Lasso framework introduced by Javanmard and Montanari (2014). The key contribution lies in reparameterizing the optimization problem used for constructing the debiasing matrix. The reformulation preserves all theoretical properties of the original debiased Lasso estimator while enabling the derivation of a simple closed-form solution for 𝑊 under a deterministic condition that holds with high probability for many random sub-Gaussian sensing matrices.

Strengths:

1. The reformulation from 𝑀 to 𝑊 elegantly simplifies the debiasing process while preserving theoretical rigor.
2. This paper provides both theoretical clarity and substantial computational acceleration.
3. The approach applies to general sub-Gaussian designs and not just to specific covariance structures.

Weaknesses:
1. The experiments, while detailed, are limited to synthetic data; real-world evaluations or robustness tests against model misspecification are absent.
2. Selecting an appropriate $\mu$ in practice may require simulation or prior knowledge of $\rho$, reducing ease of use in non-ideal settings.

**Additional Comments:**

N/A

**Audience:**

No

**Audience Explanation:**

This work is too theoretical. I am wondering if any existing works related to this topic. The baseline is from 2014, which is very old and this demonstrates that no new works have been proposed in recent years.

**Claims And Evidence:**

Yes

**Claims Explanation:**

I briefly checked the proofs and did not find any errors.

**Requested Changes:**

The authors could add more related works from recent years to demonstrate that this work is still of interest to a significant group of researchers.

Experiments on real-world data should be included.

---

### Review · Reviewer_fhgi · 2025-11-12

**Summary Of Contributions:**

\textbf{Summary of Contributions.}

This paper presents a reformulation of the classical debiased LASSO approach by Javanmard \& Montanari (2014a). Instead of computing an approximate inverse $M$ of $A^\top A/n$ via convex optimization (which is computationally expensive), the authors propose to directly compute the weight matrix $W = AM^\top$, showing that theoretical guarantees of the debiased estimator depend only on $AM^\top$, not on $M$ itself.

The key contribution is the derivation of a closed-form solution for $W$ under mild conditions on the sensing matrix $A$, particularly when the columns of $A$ are weakly correlated sub-Gaussian random vectors. The authors establish deterministic conditions for optimality (Theorem~1) and probabilistic bounds for random designs (Theorem~2), showing that the proposed solution matches the optimization-based one with high probability.

Empirical results demonstrate that the closed-form debiasing yields equivalent estimation accuracy and statistical inference to the original debiased LASSO, but with four orders of magnitude faster runtime. Extensive experiments compare sensitivity, specificity, and variance ratios across Gaussian and correlated covariance structures, consistently showing that the proposed method maintains theoretical validity while being computationally efficient.

Strengths:

1 A clean theoretical insight that reduces an iterative convex optimization problem to a closed-form expression.
2 Strong proofs of equivalence and probabilistic bounds (Theorems~1–2).
3 Solid empirical validation showing identical performance and major speed-ups.
4 Practical importance for high-dimensional inference and real-time or streaming regression settings.


Weaknesses:
\begin{itemize}
  \item The assumptions (e.g., sub-Gaussian i.i.d. design, weak correlations) may be restrictive in real-world data.
  \item The numerical experiments focus mostly on synthetic setups; no real-data illustration is provided.
\end{itemize}

**Audience:**

Yes

**Audience Explanation:**

The proposed method directly targets a fundamental limitation of high-dimensional inference—the bias of the LASSO estimator—and provides a computationally efficient debiasing technique. This topic is highly relevant to the machine learning and statistics communities, especially for researchers working on sparse estimation, convex optimization, and scalable inference. Moreover, the result has practical implications for modern high-dimensional applications (e.g., genomics, compressive sensing, causal inference), making it a strong fit for the TMLR readership.

**Broader Impact Concerns:**

The paper focuses on methodological and theoretical improvements for statistical estimation. It does not raise ethical or societal concerns. The technique improves computational efficiency for debiased estimators, which may indirectly benefit applications involving large-scale data or scientific inference. No issues of fairness, privacy, or misuse are apparent.

**Claims And Evidence:**

Yes

**Claims Explanation:**

The theoretical claims are well supported through rigorous derivations and clear proofs, including convex duality analysis and probabilistic concentration bounds. The equivalence between the original optimization and the reformulated problem is mathematically sound and verified both analytically and empirically. The experiments convincingly validate the theoretical results—the closed-form solution produces identical debiased estimates to those obtained via optimization, confirming the correctness of the formulation. Runtime analysis further supports the claim of a significant computational advantage.

Minor concerns include the narrow focus on synthetic data and the absence of robustness tests beyond Gaussian or sub-Gaussian ensembles, but these do not undermine the main theoretical claims.

**Requested Changes:**

**Discussions on choices and estimation of $\mu$**: The work mentions empirical choices for $\mu$ but does not give a prescriptive procedure for practitioners. Guidance (or an algorithm) to estimate a safe $\mu$ from data, taking into account trade-offs, would substantially increase the practical utility.

**Numerical scalability**: Experiments are informative but limited to moderate p (e.g., p=500). Showing scalability to larger
p (and reporting memory/time for both closed-form and optimization) would underline the practical speed advantage.

**Comparisons to other debiasing approaches**: The paper briefly references nodewise regression and other approaches; empirical comparison (runtime and inferential quality) to a representative nodewise-regression implementation would contextualize the gains more concretely.

---

### Review · Reviewer_sNBF · 2026-01-08

**Summary Of Contributions:**

This paper addresses the computational bottleneck of the debiased LASSO estimator, specifically the expensive optimization required to find an approximate inverse of the covariance matrix. By re-parameterizing the problem to focus on a weight matrix $W$ (where $W=AM^\top$), the authors derive a simple, closed-form solution that eliminates the need for iterative solvers, which holds when the linear correlation satisfies a condition. They provide theoretical guarantees showing that this closed-form solution is optimal under a condition on the sensing matrix's column correlations, which is satisfied with high probability for common random designs. Empirical results demonstrate that this approach maintains statistical inference quality while speeding up computation by several orders of magnitude.

**Audience:**

Yes

**Audience Explanation:**

Researchers and practitioners in the statistics community would find this paper highly relevant, particularly those focused on high-dimensional statistics, optimization, and scalable machine learning.

**Broader Impact Concerns:**

This paper focuses on developing algorithms for LASSO estimation. We do not identify any specific societal risks that need to be highlighted in this context.

**Claims And Evidence:**

Yes

**Claims Explanation:**

The authors provide solid proofs using Fenchel duality and concentration inequalities to support both the optimality and the high-probability feasibility of their solution.

**Requested Changes:**

The reviewer is not an expert on the LASSO estimator and hence can not provide a holistic evaluation of its value. However, the reviewer believes that the proposed method is interesting in itself and the analysis is basically correct. Here are some questions or comments for the paper:
1. Is it possible to provide a draft on how to derive the closed-form (7) in the main body?
2. Does Theorem 1 suggest using $\mu=\frac{\rho}{1+\rho}$ so that the proposed method consistently achieves an optimal error $\mu$?
3. It is better to add a set of experiments to compare the proposed method and the baseline in terms of both accuracy and time under different $\mu$. It would be helpful to show how we should choose different methods based on different $\mu$ requirements.

---

> ### Author Response · Authors · 2026-01-12
> **Response to Reviewer sNBF**
>
> We thank the reviewer for their review of our paper. We are responding to their concerns here below:
>
>    1) *This paper addresses the computational bottleneck of the debiased LASSO estimator, specifically the expensive optimization required to find an approximate inverse of the covariance matrix. By re-parameterizing the problem to focus on a weight matrix $\boldsymbol{W}$
>  (where $\boldsymbol{W}=\boldsymbol{AM}^{\top}$), the authors derive a simple, closed-form solution that eliminates the need for iterative solvers, which holds when the linear correlation satisfies a condition. They provide theoretical guarantees showing that this closed-form solution is optimal under a condition on the sensing matrix's column correlations, which is satisfied with high probability for common random designs. Empirical results demonstrate that this approach maintains statistical inference quality while speeding up computation by several orders of magnitude.*
>
> **Response**: We thank the reviewer for their summary and words of appreciation.
>
>  2) *Requested Changes: Is it possible to provide a draft on how to derive the closed-form (7) in the main body?*
>
>  **Response**: We thank the reviewer for their suggestion. We have now added a brief draft of the derivation of the closed-form (7) of Theorem 1 in the revised manuscript after the theorem statement.
>
>  3) *Does Theorem 1 suggest using $\mu=\frac{\rho}{1+\rho}$
>  so that the proposed method consistently achieves an optimal error $\mu$?*
>
>  **Response**: Since we know the sensing matrix $\boldsymbol{A}$, it is easy and inexpensive to compute $\rho$ as given in Theorem 1. Therefore, we can directly set $\mu = \rho/(1+\rho)$. We have now clarified this in Remark 3 after Theorem 2. We further note that we are interested in seeking smaller values of $\mu$, for which the optimization problem to obtain $\boldsymbol{W}$ remains feasible.
>
>  4) *It is better to add a set of experiments to compare the proposed method and the baseline in terms of both accuracy and time under different $\mu$. It would be helpful to show how we should choose different methods based on different $\mu$ requirements.*
>
>  **Response**: We thank the reviewer for this suggestion. We have now added a table of results in the revised manuscript. In Table 5, we present the sensitivity and specificity and the runtime for both the exact solution $\boldsymbol{W}_e$ and the optimal solution $\boldsymbol{W}_o$ for different values of $\mu$ both above and below the bound $\rho/(1+\rho)$. We see near identical sensitivity and specificity values for both debiasing matrices whereas the main difference is computational as computing $\boldsymbol{W_o}$ takes more than 550 seconds whereas $\boldsymbol{W_e}$ is obtained in a few milliseconds.

---

> > ### Comment · Reviewer_sNBF · 2026-01-19
> >
> > Thanks for the authors' detailed response and revision. All concerns have been addressed.

---

> ### Author Response · Authors · 2026-01-13
> **Response to Reviewer sNBF**
>
> We have submitted a revised version in response to the reviews of all three reviewers. Please let us know whether our submitted changes adequately address your concerns, or whether you require any further clarifications. If there are further changes needed, we still have a few days to make edits.

---

> > ### Author Response · Authors · 2026-01-17
> > **Response to reviewer sNBF**
> >
> > We have submitted a revised version in response to the reviews of all three reviewers. Please let us know whether our submitted changes adequately address your concerns, or whether you require any further clarifications. If there are further changes needed, we still have a few days to make edits.

---

### Decision · Action_Editor_nfP1 · 2026-03-02

**Recommendation:** Accept as is

**Audience:**

Yes

**Audience Explanation:**

LASSO is a broadly utilized statistical tool, and debiasing LASSO has been an important research question. The results of this paper will be interesting to the data science and statistics community.

**Claims And Evidence:**

Yes

**Claims Explanation:**

The main claim of the work is a computationally fast algorithm to find debiasing matrix M for LASSO. Authors provide theoretical guarantees that their formulation enjoys closed form solution, and that the solution satisfies the debiasing property. Authors also provide numerical results.